# THE PRICE OF EXPLAINABILITY FOR KERNEL $k$-MEANS

## ABSTRACT

The explainability of the machine learning model has received increasing attention recently for security and model reliability reasons. Recently, there has been a surge of interest in interpreting the clustering results of $k$-means and kernel $k$-means algorithms. In this paper, we study explainable kernel clustering and compare the explainable performance of kernel $k$-means algorithms based on different kernels. In particular, we show that kernel $k$-means clustering with the Laplacian kernel has lower price of explainability than that with the Gaussian kernel, which is consistent with the experimental findings of Fleissner et al. (2024). In addition, we propose a new kernel $k$-means interpretability algorithm that directly constructs a dual-threshold tree in the original space to achieve interpretable kernel $k$-means, and experimentally show that it outperforms KIMM, which constructs the threshold tree in the kernel space.

## 1 INTRODUCTION

The rapid development of machine learning has greatly facilitated our lives, especially in recent years, our ability to obtain and process data has improved dramatically Ertel (2024). Many decisions rely on the data and the algorithms to extract information. For safety (such as in the medical and autonomous driving fields) and people's trust in algorithms, we are often required to explain the results obtained by machine learning algorithms (Carvalho et al., 2019; Marcinkevičs & Vogt, 2020).

As an unsupervised learning task, clustering can mine potential groups from massive amounts of unlabeled data and provide valuable information (Aggarwal, 2015; Gan et al., 2020). As one of the most commonly used clustering algorithms, $k$-means has been widely used in many fields (Ahmed et al., 2020). Therefore, the interpretation of $k$-means has also received much attention recently (Bera et al., 2019; Bandyapadhyay et al., 2023; Dasgupta et al., 2020).

$k$-means is a partition-based clustering method that uses the EM (Expectation-maximization) algorithm (Dempster et al., 1977; Moon, 1996) to find $k$ cluster centers so that the cost based on the $k$ cluster centers is minimized, and finally, the optimal $k$ centers are obtained. A sample belongs to the cluster represented by the center closest to it. In other words, $k$-means constructs a Voronoi diagram (Aurenhammer, 1991) based on the $k$ optimal centers, in which each cell represents the area belonging to a cluster. An example with 3 clusters is shown in Figure 1(a).

However, the partitioning based on the Voronoi diagram has the following two disadvantages:

1. For the given points $\mathcal{X} \subseteq \mathbb{R}^d$, the boundary of the Voronoi diagram is not easy to represent because it is a linear combination of $d$ features. For data in the high-dimensional space, this representation will become more complicated.

2. Due to the linear combination of $d$ features on the boundary of the Voronoi diagram, when we linearly combine different features (such as height and weight), the meaning of their representation is confusing.

Based on the above two reasons, it is difficult to use the Voronoi diagram to make an acceptable and understandable explanation of the clustering results.

In order to more easily explain the clustering results of $k$-means, the threshold tree Kotsiantis (2013) is used to construct an axis-parallel partition, so that independent features are used instead of linear combinations of multiple features for explainability (Dasgupta et al., 2020). The threshold tree starts from the root node, selects a feature and threshold for partitioning each time, and recursively

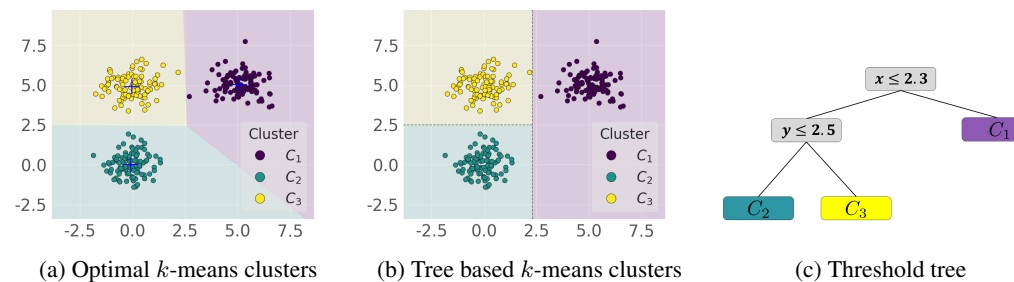

(a) Optimal $k$-means clusters    (b) Tree based $k$-means clusters    (c) Threshold tree

Figure 1: An example of explainable clustering uses the threshold tree. The optimal clustering constructs a Voronoi diagram, using multiple feature combinations. While the threshold tree uses independent features to partition points with very few mistakes.

generates a decision tree containing $k$ leaf nodes. When a decision tree is built, new points can be easily clustered to achieve explainable assignments. The threshold tree of the points shown in Figure 1(a) is built in Figure 1(b-c).

Constructing a threshold tree can produce concise and explainable clustering results, but it is at the expense of clustering accuracy. Therefore, its cost is greater than the optimal $k$-means cost. Recent studies use the optimization objective of $k$-means as a metric for evaluating thresholds (Dasgupta et al., 2020; Frost et al., 2020; Makarychev & Shan, 2021; Gamlath et al., 2021; Charikar & Hu, 2022; Esfandiari et al., 2022; Makarychev & Shan, 2022; Laber et al., 2023; Gupta et al., 2023; Makarychev & Shan, 2024; Fleissner et al., 2024), so most of them focus on constructing threshold trees at a lower cost.

For $k$-means and $k$-median, many methods have been proposed that are close to the optimal competitive ratio (Gupta et al., 2023). However, there are very few studies on kernel $k$-means (Fleissner et al., 2024). Kernel $k$-means is proposed to achieve nonlinear data clustering (Dhillon et al., 2004). It executes $k$-means in kernel space and constructs Voronoi diagrams in the corresponding reproducing kernel Hilbert space (RKHS) (Berlinet & Thomas-Agnan, 2011). The decision boundary in the original space is a very complex function (Hofmann et al., 2008), so the interpretation of kernel $k$-means is more challenging.

Kernel $k$-means has better clustering performance than $k$-means (Filippone et al., 2008), so it is very meaningful to explain the clustering results of kernel $k$-means. Inspired by explainable $k$-means that constructs a threshold tree, Fleissner et al. (2024) employs KIMM (Kernel Iterative Mistake Minimization) to construct a threshold tree in the regenerating kernel Hilbert space and then converts the threshold tree into a double threshold decision tree in the original space to explain the output of kernel $k$-means. However, this method is based on the approximation of the kernel function to obtain a finite-dimensional approximate kernel space, and the threshold tree needs to be converted from kernel space to the original Euclidean space, which sacrifices the accuracy of clustering.

In this paper, we analyze the price of explainability of kernel $k$-means based on the Gaussian kernel and Laplace kernel under different data distributions, and discuss their relationship with $k$-means and $k$-medians. We theoretically obtain the same conclusion as Fleissner et al. (2024) experimental findings, that is, the kernel $k$-means based on the Laplace kernel has a lower price of explainability than the kernel $k$-means based on the Gaussian kernel. In addition, we propose a new interpretability algorithm for kernel $k$-means, which directly builds a dual-threshold tree in the original space to achieve explainable kernel $k$-means. and experimentally show that it has better performance than KIMM.

We summarize our contributions as follows:

1. Analyzing the impact of different data distributions on the explainability of kernel $k$-means.

2. Comparing the price of explainability of the Laplacian kernel and the Gaussian kernel-based kernel $k$-means.

3. Proposing a new algorithm to build a dual-threshold tree in the original space to achieve explainable kernel $k$-means.

## 2 PRELIMINARIES AND RELATED WORK

### 2.1 $k$-MEANS AND KERNEL $k$-MEANS

Given $\mathcal{X} \subseteq \mathbb{R}^d$ sampled from distribution $\mathcal{P}$, For $k \geq 1$, a $k$-means clustering partition the dataset $\mathcal{X} = \{\mathbf{x}_1, \ldots, \mathbf{x}_n\}$ ($n = |\mathcal{X}|$) into $k$ clusters. Let $C_1, \ldots, C_k$ be the $k$ clusters obtained by a $k$-means, and $c_1, \ldots, c_k$ be the $k$ centers of each cluster. The partition formed by $k$-means clustering is a Voronoi diagram (Aurenhammer, 1991), where $c_j$ is the seed of the $j$-th cell of the Voronoi diagram, and the points $\mathbf{x}$ in the same partition with $c_j$ belong to the same cluster $C_j$. The cost of $k$-means is defined as follows:

**Definition 2.1.** The cost of a Voronoi diagram partition $\mathcal{V}$ constructed by $k$-means for $\mathcal{X}$ is:

$$cost_2(\mathcal{V}, \mathcal{X}) = \sum_{j=1}^{k} \sum_{\mathbf{x} \in C_j} \|\mathbf{x} - c_j\|_2^2,$$

where $\| \cdot \|_2$ is the $\ell_2$-norm.

Similarly, the cost of $k$-medians is defined as follows:

**Definition 2.2.** The cost of a Voronoi diagram partition $\mathcal{V}$ constructed by $k$-medians for $\mathcal{X}$ is:

$$cost_2(\mathcal{V}, \mathcal{X}) = \sum_{j=1}^{k} \sum_{\mathbf{x} \in C_j} \|\mathbf{x} - c_j\|_1,$$

where $\| \cdot \|_1$ is the $\ell_1$-norm.

Since $k$-means clustering does not work well, kernel $k$-means was proposed by using kernel-based distance instead of Euclidean distance. Kernel $k$-means uses the feature function $\psi : \mathbb{R}^d \to \mathbb{H}$ to map points $\mathcal{X}$ to $\psi(\mathcal{X}) = \{\psi(\mathbf{x}_1), \ldots, \psi(\mathbf{x}_n)\}$ in the reproducing kernel Hilbert space $\mathbb{H}$ first, and then performs $k$-means for $\psi(\mathcal{X})$. By using the kernel trick $\kappa(\mathbf{x}, \mathbf{y}) = \psi(\mathbf{x})^\top \psi(\mathbf{y})$, the calculation of $\psi(\cdot)$ can be avoided. The cost of kernel $k$-means using kernel $\kappa$ is defined as follows:

**Definition 2.3.** Let $\mu_1, \ldots, \mu_k$ be the centers of $C_1, \ldots, C_k$ in $\mathbb{H}$, The cost of a Voronoi diagram partition $\mathcal{V} \in \mathbb{H}$ constructed by kernel $k$-means based on kernel $\kappa$ for $\mathcal{X}$ is:

$$J(\mathcal{V}, \mathcal{X}) = \sum_{j=1}^{k} \sum_{\mathbf{x} \in C_j} \|\psi_\kappa(\mathbf{x}) - \mu_j\|^2 \tag{1}$$

### 2.2 EXPLAINABILITY OF $k$-MEANS AND KERNEL $k$-MEANS

Voronoi diagrams are challenging to interpret, which has led researchers in recent years to explore the use of threshold trees as a means to clarify the outcomes of $k$-means clustering. While threshold trees enhance the interpretability of $k$-means results, they may come at the cost of reduced clustering performance. This presents a trade-off between the clarity of interpretation and the efficiency of clustering. Over the past few years, many methods have been proposed to strike a balance, aiming to preserve strong clustering performance alongside interpretability. The effectiveness of interpretable $k$-means clustering is often gauged by what is termed the 'price of explainability'.

**Definition 2.4.** For a dataset $\mathcal{X}$ and an interpretable threshold tree $\mathcal{T}$, the price of explainability of $\mathcal{T}$ is defined as the ratio $\varrho$:

$$\varrho(\mathcal{T}, \mathcal{X}) = \frac{cost(\mathcal{T}, \mathcal{X})}{\min_\mathcal{V} cost(\mathcal{V}, \mathcal{X})},$$

where $cost(\cdot)$ is the cost of the clustering.[1] The price of explainability of $\mathcal{X}$ is $\varrho(\mathcal{X}) = \min_\mathcal{T}(\mathcal{T}, \mathcal{X})$.

---

[1] $cost(\mathcal{T}, \mathcal{X})$ is the cost of explainability.

As one of the earliest studies on the interpretability of $k$-means and $k$-medians, Dasgupta et al. (2020) proposed using threshold trees to explain the clustering results, and developed Iterative Mistake Minimization (IMM) algorithms for constructing threshold trees with a price of $O(k)$ for $k$-medians and a price of $O(k^2)$ for $k$-means. (Makarychev & Shan, 2021) randomly selects cut to achieve a price of $O(\log k \log k \log k)$ for $k$-medians and a price of $O(k \log k \log k \log k)$ for $k$-means, while IMM greedily selects the cut with the least error each time to build a threshold tree. Laber & Murtinho (2021) further improves the price for low-dimensional space, shows a price of $O(d \log k)$ for $k$-medians and $O(kd \log k)$ for $k$-means. As more and more methods are proposed(Gamlath et al., 2021; Charikar & Hu, 2022; Esfandiari et al., 2022; Fleissner et al., 2024; Gupta et al., 2023; Makarychev & Shan, 2024; 2022), the upper bound of the price of explainability of $k$-medians is gradually reduced to $1 + H_{k+1}$, and the lower bound is $(1 - o(1)) \ln k$, where $H_{k-1} = \frac{1}{1} + \frac{1}{2} + \cdots + \frac{1}{k-1}$ is the $(k-1)^{th}$ harmonic number. The upper bound of the price of explainability of $k$-means is gradually reduced to $O(k \ln \ln k)$, and the lower bound is $\Omega(k)$.

Fleissner et al. (2024) extended the realm of interpretability to kernel $k$-means, a method known for its superior performance to $k$-means. Achieving interpretability in this context presents unique challenges, as kernel $k$-means operates within the RKHS $\mathbb{H}$, whereas interpretability is sought in the original feature space where threshold trees are constructed. To bridge this gap, Fleissner et al. (2024) employed such as kernel-based Iterative Mistake Minimization (KIMM) to initially develop a threshold tree in RKHS $\mathbb{H}$, followed by translating these thresholds back into the original space $\mathbb{R}^d$. However, since IMM is inherently limited to finite-dimensional spaces, they have to approximate the kernel function using the finite-dimensional features. This approach, while facilitating the interpretability process, inevitably introduces a trade-off with accuracy.

## 3 THE PRICE OF KERNEL $k$-MEANS

The No Free Lunch (NFL) theorem (Zhou & Liu, 2021) implies that no single algorithm can universally outperform all others across every possible scenario. In the context of clustering, this means that neither $k$-means nor kernel $k$-means can be deemed universally superior. Instead, their performance is highly dependent on the specific characteristics of the data distribution. In this section, we will delve into various data distributions and explore how they influence the explainability of kernel $k$-means.

### 3.1 LINEAR KERNEL BASED KERNEL $k$-MEANS

In addition to the data distribution, the clustering performance of kernel $k$-means is also related to the choice of kernel function (Dhillon et al., 2004; Ahmed et al., 2020). Different kernels map data from $\mathbb{R}^d$ to different Reproducing Kernel Hilbert Spaces (RKHS), resulting in varied data distributions in the RKHS. Consequently, the clustering results obtained using $k$-means will differ. Similarly, different kernels map data to different RKHS spaces, which also presents varying challenges to the explainability of the clustering results. In this subsection, we will analyze the simple linear kernel:

$$\kappa(\mathbf{x}, \mathbf{y}) = \mathbf{x}^\top \mathbf{y}.$$

Since the $k$ centers are pivotal for the interpretability of $k$-means clustering, they likewise underpin the interpretability of kernel $k$-means clustering results. We first define the bijective centers:

**Definition 3.1.** (Bijective center) If for distribution $\mathcal{P}$, $\mathcal{X} \subseteq \mathbb{R}^d$ is drawn from $\mathcal{P}$, for kernel $\kappa$, there exists a point $\wp \in \mathbb{R}^d$ such that

$$\psi(\wp) = \frac{1}{|\mathcal{X}|} \sum_{\mathbf{x} \in \mathcal{X}} \psi(\mathbf{x}) = \mu \in \mathbb{H},$$

then $\mu$ is called a bijective center, where $\mathbb{H}$ is RKHS.

**Theorem 3.2.** *When $\kappa(\cdot, \cdot)$ is a linear kernel, $\forall \mathcal{P}$, $\exists \wp \in \mathbb{R}^d$, $\psi(\wp)$ is a bijective center, and*

$$\wp = c = \frac{1}{|\mathcal{X}|} \sum_{x \in \mathcal{X}} x.$$

Theorem 3.2 makes sense intuitively since the data distribution before and after the linear kernel mapping is consistent. Figure 2 shows an example of a Gaussian distribution (left) and an example of data uniformly distributed on a circle (middle).

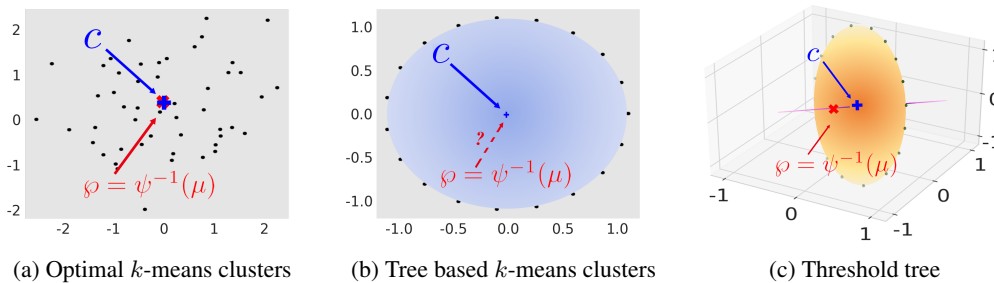

(a) Optimal $k$-means clusters  (b) Tree based $k$-means clusters  (c) Threshold tree

Figure 2: An illustration of center $c$ and the image $\wp$ of bijective center $\mu$.

The linear kernel is the simplest and most intuitive kernel function, which directly calculates the inner product of two vectors in the original input space. When we employ the linear kernel in kernel $k$-means, we have the following theorem:

**Theorem 3.3.** *The price of explainability for kernel $k$-means employing linear kernel is $O(k \ln \ln k)$.*

Theorem 3.3 shows the low price of explainability of linear kernel-based kernel $k$-kmeans. Since the mapping function of the linear kernel is $\psi(\mathbf{x}) = \mathbf{x}$, the distribution of the mapped data is consistent with the original space. For all clusters $C_i$ obtained by kernel $k$-means, the cluster center $\mu_i \in \mathbb{H}$ is the bijective center, and $\mu_i = c_i$. so the existing explainable $k$-means methods can be used directly.

### 3.2 GAUSSIAN KERNEL BASED KERNEL $k$-MEANS

However, when employing kernels such as the Gaussian, which is defined as: $\kappa(\mathbf{x}, \mathbf{y}) = \exp\left(-\frac{\|\mathbf{x}-\mathbf{y}\|_2^2}{2\sigma^2}\right)$ where $\sigma > 0$ is the bandwidth parameter of the kernel function. The existence of $\wp$ for distribution $\mathcal{P}$ is not guaranteed for the bijective center. For instance, in the case of the simple distribution depicted in the middle of Figure 2, $\wp$ does not exist. Conversely, if the circle is situated in $\mathbb{R}^3$ space, then $\wp$ does exist for bijective center $\mu = \psi(\wp)$, specifically on the line that passes through the center of the circle and is perpendicular to the plane containing the circle [2].

**Theorem 3.4.** *Given data $\mathcal{X} \in \mathbb{R}^d$, if exist bijective centers $\wp \in \mathbb{R}^d$, the price of explainability for kernel $k$-means employing Gaussian kernel is $O(k \ln \ln k)$.*

Theorem 3.4 shows that kernel $k$-means with a Gaussian kernel has the same price of explainability as kernel $k$-means with a linear kernel. This is because the data distribution is simple if $\wp \in \mathbb{R}^d$, such as the simple example given in Figure 2c. Therefore, its explainability is not that difficult.

#### 3.2.1 ORDER-PRESERVING POINTS EXIST IN THE DATA DISTRIBUTION

Theorem 3.4 shows that when $\wp \in \mathbb{R}^d$, the price of explainability of the kernel $k$-mean algorithm based on the Gaussian kernel is low, but in the real world, the data distribution is often more complex. In the following two subsections, we will analyze the price of explainability of the kernel $k$-means based on the Gaussian kernel under more complex data distributions. Step by step, we will analyze a less complex distribution in this subsection and introduce the more complex case in the next subsection.

**Definition 3.5.** (Order-preserving points) Given $\mathcal{X} = \{C_1, C_2, \ldots, C_k\} \in \mathbb{R}^d$, $\kappa$ be a kernel. $\forall x \in \mathcal{X}$, and let $\psi(x)$ denote the mapping. Define $\mu_1, \mu_2, \ldots, \mu_k$ as the centers of the $C_1, C_2, \ldots, C_k$ in the

---

[2]The proof is given in Appendix.

space $\mathbb{H}$ induced by $\kappa$: $\mu_i = \frac{1}{|C_i|} \sum_{x \in C_i} \psi(x), i = \{1, 2, \ldots, k\}$. We say that $\wp_1, \wp_2, \ldots, \wp_k \in \mathbb{R}^d$ are order-preserving points if:

$$\forall x \in \mathcal{X}, \forall i, j \in \{1, 2, \ldots, k\}, i \neq j,$$
$$\|\psi(x) - \mu_i\| \leq \|\psi(x) - \mu_j\| \iff \|x - \wp_i\|_l \leq \|x - \wp_j\|_l.$$

where $\| \cdot \|$ denotes the $\ell_2$ norm in the spaces $\mathbb{H}$, and $\| \cdot \|_l$ denotes the $\ell_l$ norm in the spaces $\mathbb{R}^d$.

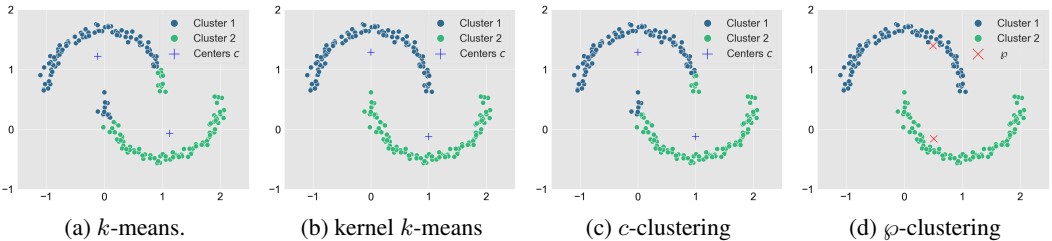

(a) $k$-means.  (b) kernel $k$-means  (c) $c$-clustering  (d) $\wp$-clustering

Figure 3: An example of $k$-means, kernel $k$-means, centers $c$ and the order-preserving points $\wp$. (a) The clustering result obtained by $k$-means. (b) The clustering result obtained by kernel $k$-means. (c) The clustering result obtained by the centers $c_i$ ($c_i$ are the centers of $C_i$ obtained by kernel $k$-means as shown in (b))in $\mathbb{R}^d$. (d) The clustering result obtained by order-preserving points $\wp_i \in \mathbb{R}^d$.

According to Definition 3.5, if there are order-preserving points $\wp_1, \ldots, \wp_k \in \mathbb{R}^d$, then we will obtain the same clustering result as that obtained by using the centers $\mu_1, \ldots, \mu_k$ through partitioning in $\mathbb{H}$, by employing the order-preserving points $\wp_1, \wp_2, \ldots, \wp_k$ for partitioning in $\mathbb{R}^d$.

Figure 3 shows an example of order-preserving points. $k$-means can not obtain a perfect clustering result as shown in Figure 3(a), while kernel $k$-means perform well (construct a Voronoi diagram in $\mathbb{H}$) on this data (Figure 3(b)). even if we use the centers $c_1, \ldots, c_k$ of $C_1, \ldots, C_k$ obtained by the kernel $k$-means to construct the partition (Voronoi diagram in $\mathbb{R}^d$), only a slightly better clustering result than $k$-means can be obtained, still can not cluster all points correctly. However, the partitions (Voronoi diagram in $\mathbb{R}^d$) constructed by order-preserving points $\wp_1, \wp_2, \ldots, \wp_k$ can correctly cluster all data points.

Why can not $k$-means get a good clustering result, but order-preserving points can when both constructing a Voronoi diagram in $\mathbb{R}^d$ space? This is because $k$-means is based on the EM algorithm, and order-preserving points may not be in the solution space of $k$-means. Because the seed in the Voronoi diagram in the EM algorithm is the center of a subset of $\mathcal{X}$, order-preserving points may not be obtained from a subset of $\mathcal{X}$.

**Theorem 3.6.** *Given data $\mathcal{X} \in \mathbb{R}^d$, if exist order-preserving points $\wp_1, \ldots, \wp_k \in \mathbb{R}^d$, the price of explainability for kernel $k$-means employing Gaussian kernel is $O(k \ln \ln k)$.*

Although the distribution becomes more complicated ($k$-means cannot get good clustering results), Theorem 3.6 shows that the kernel $k$-means algorithm based on the Gaussian kernel still has the same price of explainability as in Theorem 3.4. This is because although $k$-means cannot find a Voronoi diagram constructed with $k$ centers using the EM algorithm so that all points are correctly clustered, such points (order-preserving points) do exist in the $\mathbb{R}^d$ space.

### 3.2.2 ORDER-PRESERVING POINTS ARE NON-EXISTENT IN THE DATA DISTRIBUTION

The premise for kernel $k$-means to obtain a low price of explainability $O(k \ln \ln k)$ is that a Voronoi diagram in the $\mathbb{R}^d$ space can perfectly divide all points. Such as the Voronoi diagram constructed by the seeds as bijective centers and order-preserving points. What is the price of explainability if such a Voronoi diagram does not exist?

Figure 4 shows an example of a data distribution where the order-preserving point does not exist. $k$-means achieves very poor clustering results on this dataset, while kernel $k$-means achieves perfect clustering results. At this time, if a single threshold tree is used, very poor clustering results will be obtained. Therefore, a dual-threshold tree is needed for obtaining a good clustering result, which is shown in Figure 4(c). We will discuss it in the next section.

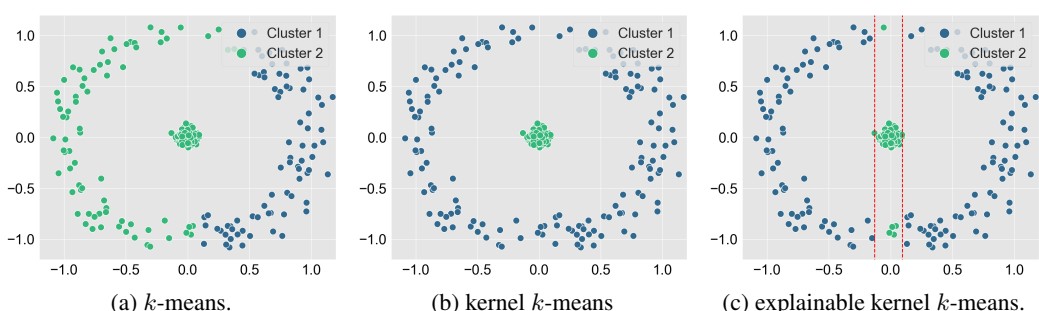

Figure 4: An example of $k$-means, kernel $k$-means, and explainable kernel $k$-means on a dataset that order-preserving points are non-existent using DT$^2$ introduced in Section 4.

**Theorem 3.7.** *Given data* $\mathcal{X} \in \mathbb{R}^d$, *if order-preserving points are non-existent, the cost of explainability for kernel $k$-means employing Gaussian kernel is at most* $(1 - e^{-\gamma R})n + \frac{e^{-\gamma R}}{2}O(k \ln \ln k) \cdot cost_{opt(\mathcal{X})}$, *where $R$ is the radius of dataset $\mathcal{X}$, $cost_{opt(\mathcal{X})}$ is the optimal kernel $k$-means cost.*

**Theorem 3.8.** *Given data* $\mathcal{X} \in \mathbb{R}^d$, *if order-preserving points are non-existent, the cost of explainability for kernel $k$-means employing Gaussian kernel is at least* $(1 - e^{\gamma R})n + \frac{e^{\gamma R}\Omega(k)}{2} \cdot cost_{opt(\mathcal{X})}$, *where $R$ is the radius of dataset $\mathcal{X}$, $cost_{opt(\mathcal{X})}$ is the optimal kernel $k$-means cost.*

Theorem 3.7 and 3.8 show the upper bound and lower bound of the cost of explainability of kernel $k$-means based on the Gaussian kernel, the price of explainability is related to the distribution of the data (the factor $R$). If the data distribution is more complex, there will be a higher price.

### 3.3 LAPLACIAN KERNEL BASED KERNEL $k$-MEANS

Another widely used kernel function is the Laplacian kernel, which is defined as $\kappa(\mathbf{x}, \mathbf{y}) = \exp\left(-\frac{\|\mathbf{x}-\mathbf{y}\|_1^2}{2\sigma_1^2}\right)$. Corresponding to the three cases of the Gaussian kernel, we give the price of explainability of the kernel $k$-means algorithm based on the Laplacian kernel as follows:

**Theorem 3.9.** *Given data* $\mathcal{X} \in \mathbb{R}^d$, *if exist bijective centers $\wp \in \mathbb{R}^d$, the price of explainability for kernel $k$-means employing Laplacian kernel is* $2(\ln k + 1)$.

**Theorem 3.10.** *Given data* $\mathcal{X} \in \mathbb{R}^d$, *if exist order-preserving points $\wp_1, \ldots, \wp_k \in \mathbb{R}^d$, the price of explainability for kernel $k$-means employing Laplacian kernel is* $2(\ln k + 1)$.

**Theorem 3.11.** *Given data* $\mathcal{X} \in \mathbb{R}^d$, *if order-preserving points are non-existent, the cost of explainability for kernel $k$-means employing Laplacian kernel is at most* $(1 - e^{-\gamma R})n + \frac{1}{2}e^{-\gamma R}(1 - o(1)) \ln k \cdot cost_{opt(\mathcal{X})}$, *where $R$ is the $R$ is the radius of dataset $\mathcal{X}$, $cost_{opt(\mathcal{X})}$ is the optimal kernel $k$-means cost.*

**Theorem 3.12.** *Given data* $\mathcal{X} \in \mathbb{R}^d$, *if order-preserving points are non-existent, the price of explainability for kernel $k$-means employing Laplacian kernel is at least* $(1 - e^{\gamma R})n + \frac{1}{2}e^{\gamma R}(1 + o(1)) \ln k \cdot cost_{opt(\mathcal{X})}$, *where $R$ is the $R$ is the radius of dataset $\mathcal{X}$, $cost_{opt(\mathcal{X})}$ is the optimal kernel $k$-means cost.*

By observing Theorems 3.4 - 3.12, we find that the Laplacian kernel-based $k$-means has a lower price of explainability than the Gaussian kernel-based kernel $k$-means, which is consistent with the experimental findings of Fleissner et al. (2024), who pointed out in their paper that *"we observe in practice that the surrogate features defined in (4) perform well, sometimes even outperforming the surrogate Taylor features of the Gaussian kernel"*.

Specifically, the price of explainability of the kernel $k$-means based on the Gaussian kernel is controlled by '$k \ln \ln k$', while the price of explainability of the $k$-means based on the Laplacian kernel is controlled by '$\ln k$'. We find that this is because the price of explainability of the Gaussian kernel is related to the price of $k$-means, while the Laplacian kernel is related to the $k$-median. Because the Gaussian kernel is the similarity based on the 2-norm, it establishes a close relationship

with $k$-means, which is also based on the 2-norm. Conversely, the Laplacian kernel, derived from the similarity of the 1-norm, forms a strong connection with $k$-medians, which relies on the 1-norm. This discovery will provide new insights into the connection between the price of explainability of $k$-means and kernel $k$-means, as well as new interpretable algorithms. This also suggests that we use 1-norm-based kernels instead of 2-norm-based kernels when we need an interpretable kernel $k$-means for a lower price of explainability.

## 4 DUAL-THRESHOLD TREE AND EXPERIMENTS

Inspired by kernel IMM (KIMM) (Fleissner et al., 2024) (performed the IMM algorithm in $\mathbb{H}$ space and mapped the thresholds back to $\mathbb{R}^d$ space to build a tree determined by two thresholds), we construct the Dual-Threshold Tree ($DT^2$) directly in the $\mathbb{R}^d$ space, which does not require approximation of the kernel function like KIMM, thus having a lower price of explainability. The algorithm [3] is shown in Algorithm $DT^2$:, $DT^2$ uses a greedy method to select the partition with the lowest cost each time and uses a dual threshold to divide the data into the left and right nodes, until each leaf node contains only one cluster, and finally a tree with $k$ leaf nodes is constructed.

---

**Algorithm $DT^2$:** Dual-Threshold Tree for interpretable kernel $k$-means

**Input:** Data set $\mathcal{X}$, number of clusters $k$, kernel $\kappa$
**Output:** Interpretable Dual - Threshold Tree $\mathcal{T}$ with $k$ leaves

1 Perform ($\kappa$-) kernel $k$-means clustering on $\mathcal{X}$ to obtain $C = \{C_1, C_2, \ldots, C_k\}$
2 **return** $\mathcal{T} \leftarrow$ build_$DT^2(\mathcal{X}, C, \kappa)$

**build_$DT^2$** $(\mathcal{X}, C, \kappa)$:
  1    **if** *only one cluster $y$ in $\mathcal{X}$* **then**
  2      $\lfloor$ **return** $leaf.cluster \leftarrow y \in C$
  3    $(i, \theta_1, \theta_2, C) \leftarrow$
      $\min_{i, \min \mathcal{X}^i \leq \theta_1 \leq \theta_2 \leq \max \mathcal{X}^i}$ cost_split$(\mathcal{X}, C, \kappa, i, \theta_1, \theta_2)$
  4    $(\mathcal{X}_l, C_l) \leftarrow \{(\mathbf{x} \in \mathcal{X}, y \in C$ is the label of $\mathbf{x}) \mid \mathbf{x}^i \in [\theta_1, \theta_2]\}$
  5    $(\mathcal{X}_r, C_r) \leftarrow \{(\mathbf{x} \in \mathcal{X}, y \in C$ is the label of $\mathbf{x}) \mid \mathbf{x}^i \notin [\theta_1, \theta_2]\}$
  6    $node.l \leftarrow$ build_$DT^2(\mathcal{X}_l, C_l, \kappa)$
  7    $node.r \leftarrow$ build_$DT^2(\mathcal{X}_r, C_r, \kappa)$
  8    **return** node

**cost_split** $(\mathcal{X}, C, \kappa, i, \theta_1, \theta_2)$:
  1    $k \leftarrow$ the number of clusters in $C$
  2    $(\mathcal{X}_l, C_l) \leftarrow \{(\mathbf{x} \in \mathcal{X}, y \in C$ is the label of $\mathbf{x}) \mid \mathbf{x}^i \in [\theta_1, \theta_2]\}$
  3    $(\mathcal{X}_r, C_r) \leftarrow \{(\mathbf{x} \in \mathcal{X}, y \in C$ is the label of $\mathbf{x}) \mid \mathbf{x}^i \notin [\theta_1, \theta_2]\}$
  4    **for** $j \in [1, \ldots, k]$ **do**
  5      **if** $|C_{lj}| \geq |C_{rj}|$ **then**
  6        $\lfloor$ $C_{\mathbf{x}} \leftarrow \{-1 | C_{\mathbf{x}} \in C_{rj}\}$
  7      **else**
  8        $\lfloor$ $C_{\mathbf{x}} \leftarrow \{-1 | C_{\mathbf{x}} \in C_{lj}\}$
  9    $C \leftarrow$ Assign all points with label -1 to the nearest cluster
  10    $cost \leftarrow$ compute cost using Equation 1 with $C$
  11    **return** cost

---

We compared $DT^2$ with KIMM in the 5 datasets *Pahtbased* (Chang & Yeung, 2008), *Aggregaion* (Gionis et al., 2007), *Flame* (Fu & Medico, 2007), *Iris* (Fisher, 1936) and *Cancer* (Wolberg & Street, 1993) used in Fleissner et al. (2024). As in previous work, we select the bandwidth $\sigma \in \{0.01, 0.05, 0.1, 0.5, 1, 5, 10\}$ corresponding to the best clustering result in terms of ARI (adjusted rand score) (Pedregosa et al., 2011) and report the price under the best $\sigma$. The results are shown in

---

[3]The code is available in `https://anonymous.4open.science/r/DT2/`

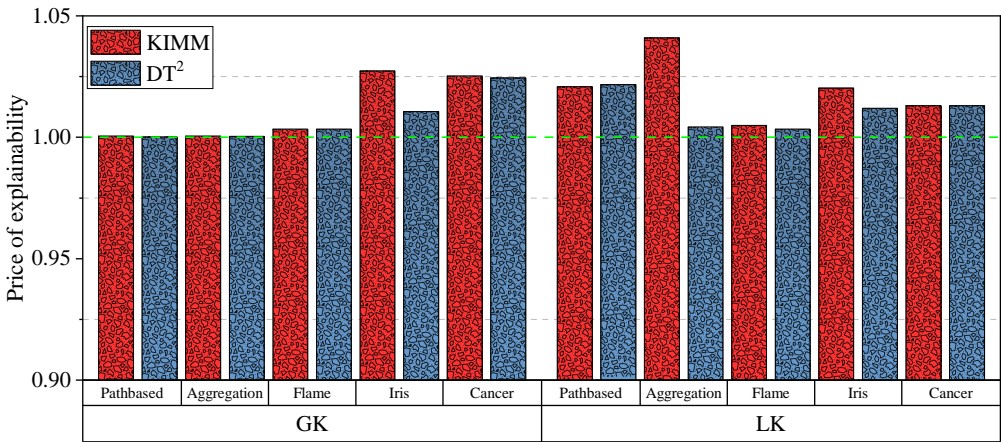

Figure 5: Comparison of the price of explainability between DT$^2$ and KIMM for (Gaussian and Laplacian) kernel $k$-means.

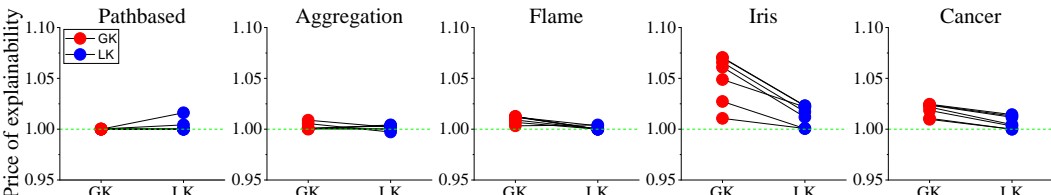

Figure 6: Comparison of the price of explainability between Gaussian and Laplacian kernel-based kernel $k$-means with various bandwidth $\sigma$. Two points connected by a bar have the same bandwidth.

Figure 5. DT$^2$ and KIMM have low price of explainability on the five datasets. DT$^2$ has a lower price of explainability than KIMM, especially on the *Iris* and *Aggregation* datasets when the Laplacian kernel is used. There are two main reasons: i) DT$^2$ does not need to approximate the kernel. ii) DT$^2$ builds the tree directly in $\mathbb{R}^d$ space, while KIMM builds the tree in $\mathbb{H}$ and then maps it to $\mathbb{R}^d$ space, which also produces more errors.

We compare the impact of different kernels on the price of explainability of the kernel $k$-means algorithm in Figure 6. Except for *Pashbased* dataset, the kernel $k$-means based on Laplacian kernel has lower prices of explainability than the Gaussian kernel-based kernel $k$-means algorithm on the other four datasets. This is consistent with our theorem and the experimental findings of Fleissner et al. (2024).

## 5 CONCLUSION

In this paper, we study the interpretability of the kernel $k$-means algorithm, analyze the impact of different kernels (such as the Gaussian kernel and the Laplacian kernel) on the price of explainability of kernel $k$-means, and propose a new interpretability algorithm DT$^2$. We find that the Laplacian kernel-based kernel $k$-means outperforms the Gaussian kernel-based kernel $k$-means in terms of the price of explainability, which is consistent with our theoretical analysis and experimental results, and consistent with the finding of Fleissner et al. (2024). At the same time, we find a close connection between Gaussian kernel-based kernel $k$-means and $k$-means, as well as between Laplacian kernel-based kernel $k$-means and $k$-medians. These findings provide new insights into the interpretability research of kernel $k$-means and lay the foundation for the development of more efficient interpretable kernel $k$-means in the future. In addition, the DT$^2$ algorithm avoids the approximation of the kernel function by directly constructing a dual-threshold tree in the original space, thereby showing a lower price of explainability than KIMM on multiple datasets.

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

## A    LARGE LANGUAGE MODELS

We used Large Language Models to polish our writing.

## B    LIMITATION

We give the price of explainability of kernel $k$-means based on linear kernels, Gaussian kernels, and Laplacian kernels. The price of explainability of other kernels (such as neural tangent kernels, etc.) and tighter bounds in the case of order-preserving points are non-existent are still problems that need to be solved.

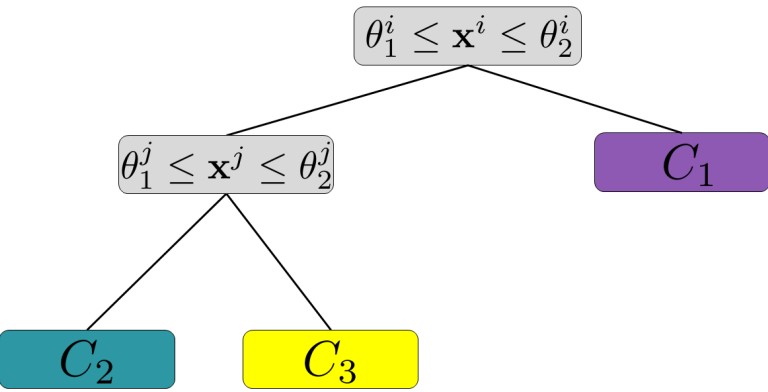

Figure 7: An illustration of the tree constructed by DT$^2$.

Table 1: The best bandwidth for the Gaussian kernel (GK) and the Laplacian kernel (LK).

|    | Pathbased | Aggregation | Flame | Iris | Cancer |
|----|-----------|-------------|-------|------|--------|
| GK | 10        | 0.1         | 5     | 10   | 0.01   |
| LK | 10        | 0.05        | 10    | 1    | 0.05   |

## C    EXPERIMENT DETAILS

The experiments are executed on a Linux machine with 1TB RAM and an AMD 128-core CPU, with each core running at 2 GHz.

The Pathbased, Aggregation, and Flame datasets are synthetic datasets. Iris contains 150 instances with 4 features, and contains 3 types of iris plant with 50 instances each. One class is linearly separable from the other 2; the latter are not linearly separable from each other. Cancer contains 569 instances with 30 features, which describe characteristics of the cell nuclei present in the image.

The source code used in the experiment:

1. KIMM: `https://github.com/maxf14/explaining_kernel_clustering`
2. DT$^2$: `https://anonymous.4open.science/r/DT2/`

The DT$^2$ constructs an interpretable tree as shown in Figure 7.

The best bandwidths used are shown in Table 1.

# D SIMPLIFICATION OF THE COST FUNCTION FOR KERNEL $k$-MEANS

The simplified cost function for kernel $k$-means is already available, and we list it here for ease of reading.

The cost function $J$ for Kernel $k$-Means is initially given by:

$$J = \sum_{j=1}^{k} \sum_{\mathbf{x} \in C_j} \left( K(\mathbf{x}, \mathbf{x}) - 2 \frac{1}{|C_j|} \sum_{\mathbf{x}' \in C_j} K(\mathbf{x}, \mathbf{x}') + \frac{1}{|C_j|^2} \sum_{\mathbf{x}', \mathbf{x}'' \in C_j} K(\mathbf{x}', \mathbf{x}'') \right)$$

Break it down into three separate sums:

The sum of the kernel evaluations of each point with itself: $\sum_{j=1}^{k} \sum_{\mathbf{x} \in C_j} K(\mathbf{x}, \mathbf{x})$

The sum of the kernel evaluations between each point and all other points in the same cluster averaged over the cluster size: $-2 \sum_{j=1}^{k} \frac{1}{|C_j|} \sum_{\mathbf{x} \in C_j} \sum_{\mathbf{x}' \in C_j} K(\mathbf{x}, \mathbf{x}')$

The sum of the kernel evaluations between all pairs of points in the same cluster, divided by the square of the cluster size: $\sum_{j=1}^{k} \frac{1}{|C_j|^2} \sum_{\mathbf{x}', \mathbf{x}'' \in C_j} K(\mathbf{x}', \mathbf{x}'')$

Combining these terms, we get:

$$J = \sum_{j=1}^{k} \sum_{\mathbf{x} \in C_j} K(\mathbf{x}, \mathbf{x}) - 2 \sum_{j=1}^{k} \frac{1}{|C_j|} \sum_{\mathbf{x} \in C_j} \sum_{\mathbf{x}' \in C_j} K(\mathbf{x}, \mathbf{x}') + \sum_{j=1}^{k} \frac{1}{|C_j|} \sum_{\mathbf{x}', \mathbf{x}'' \in C_j} K(\mathbf{x}', \mathbf{x}'')$$

Notice that the second and third terms can be combined since they both involve sums over pairs of points in the same cluster. This leads to the simplified form of the cost function:

$$J = \sum_{j=1}^{k} \sum_{\mathbf{x} \in C_j} K(\mathbf{x}, \mathbf{x}) - \sum_{j=1}^{k} \frac{1}{|C_j|} \sum_{\mathbf{x} \in C_j} \sum_{\mathbf{x}' \in C_j} K(\mathbf{x}, \mathbf{x}')$$

This simplified form highlights the key components of the cost function: the total kernel evaluations of each point with itself, and the average kernel evaluations between each point and all other points within the same cluster.

# E BIJECTIVE CENTERS

## E.1 EXAMPLE 1: LINEAR KERNEL

The linear kernel function is defined for two vectors $\mathbf{x}, \mathbf{y} \in \mathbb{R}^d$ as: $\kappa(\mathbf{x}, \mathbf{y}) = \mathbf{x}^\top \mathbf{y}$

Let's consider sampling from a probability distribution $\mathcal{P}$ over $\mathbb{R}^d$ and getting a set of independent and identically distributed samples $\mathcal{X} = \{\mathbf{x}_1, \mathbf{x}_2, \ldots, \mathbf{x}_n\}$. The $\psi(c)$ in the kernel space is computed as: $\psi(c) = \mathbb{E}_{\mathbf{x} \sim \mathcal{P}}[\kappa(\mathbf{x}, c)] = \mathbb{E}_{\mathbf{x} \sim \mathcal{P}}[\mathbf{x}^\top c]$

By the linearity of expectation, we can rewrite it as follows. $\psi(c) = c^\top \mathbb{E}_{\mathbf{x} \sim \mathcal{P}}[\mathbf{x}]$ Here, $\mathbb{E}_{\mathbf{x} \sim \mathcal{P}}[\mathbf{x}]$ is the expectation of $\mathbf{x}$ under the probability distribution $\mathcal{P}$, and $c$ is precisely the corresponding point in the original space $S$ for which $\psi(c)$ is defined.

## E.2 EXAMPLE 2: ONE-DIMENSIONAL GAUSSIAN DISTRIBUTION

**Theorem E.1.** *The one-dimensional Gaussian distribution with the Gaussian kernel does not have bijective centers in $\mathbb{R}$.*

*Proof.* Let's consider the Gaussian kernel function defined as: $\kappa(\mathbf{x}, \mathbf{y}) = \exp\left(-\frac{\|\mathbf{x}-\mathbf{y}\|^2}{2\sigma^2}\right)$ where $\sigma > 0$ is the bandwidth parameter of the kernel function.

The probability density function (PDF) of a one-dimensional Gaussian distribution is: $p(\mathbf{x}) = \frac{1}{\sqrt{2\pi\sigma_p^2}} \exp\left(-\frac{(\mathbf{x}-\mu)^2}{2\sigma_p^2}\right)$ where $\mu$ is the mean and $\sigma_p > 0$ is the standard deviation.

The kernel mean embedding $\mu_{\mathcal{P}}$ for a given probability distribution $\mathcal{P}$ is defined in the reproducing kernel Hilbert space (RKHS) as: $\mu_{\mathcal{P}} = \int_{\mathbb{R}} \kappa(\cdot, \mathbf{x}) \, d\mathcal{P}(\mathbf{x})$ For the Gaussian distribution $\mathcal{P}$, this can be written as: $\mu_{\mathcal{P}}(\mathbf{x}) = \int_{\mathbb{R}} \exp\left(-\frac{(\mathbf{x}-\mathbf{y})^2}{2\sigma^2}\right) \cdot \frac{1}{\sqrt{2\pi\sigma_p^2}} \exp\left(-\frac{(\mathbf{y}-\mu)^2}{2\sigma_p^2}\right) \, d\mathbf{y}$.

Combining the two exponential terms: $\mu_{\mathcal{P}}(\mathbf{x}) = \frac{1}{\sqrt{2\pi\sigma_p^2}} \int_{\mathbb{R}} \exp\left(-\frac{(\mathbf{x}-\mathbf{y})^2}{2\sigma^2} - \frac{(\mathbf{y}-\mu)^2}{2\sigma_p^2}\right) \, d\mathbf{y}$

Combining the exponential terms: $-\frac{(\mathbf{x}-\mathbf{y})^2}{2\sigma^2} - \frac{(\mathbf{y}-\mu)^2}{2\sigma_p^2} = -\frac{1}{2}\left(\frac{(\mathbf{x}-\mathbf{y})^2}{\sigma^2} + \frac{(\mathbf{y}-\mu)^2}{\sigma_p^2}\right)$

Complete the square:

$$\frac{(\mathbf{x}-\mathbf{y})^2}{\sigma^2} + \frac{(\mathbf{y}-\mu)^2}{\sigma_p^2} = \frac{\sigma_p^2(\mathbf{x}-\mathbf{y})^2 + \sigma^2(\mathbf{y}-\mu)^2}{\sigma^2\sigma_p^2}$$

$$= \frac{\sigma_p^2\mathbf{x}^2 - 2\sigma_p^2\mathbf{xy} + \sigma_p^2\mathbf{y}^2 + \sigma^2\mathbf{y}^2 - 2\sigma^2\mathbf{y}\mu + \sigma^2\mu^2}{\sigma^2\sigma_p^2}$$

$$= \frac{(\sigma_p^2 + \sigma^2)\mathbf{y}^2 - 2(\sigma_p^2\mathbf{z} + \sigma^2\mu)\mathbf{y} + (\sigma_p^2\mathbf{x}^2 + \sigma^2\mu^2)}{\sigma^2\sigma_p^2}$$

Let $A = \sigma_p^2 + \sigma^2$, $B = \sigma_p^2\mathbf{x} + \sigma^2\mu$, then the equation is: $\frac{A\mathbf{y}^2 - 2B\mathbf{y} + C}{\sigma^2\sigma_p^2}$ where $C = \sigma_p^2\mathbf{x}^2 + \sigma^2\mu^2$.

Completing the square:

$$\frac{A\left(\mathbf{y}^2 - \frac{2B}{A}\mathbf{y} + \left(\frac{B}{A}\right)^2\right) - A\left(\frac{B}{A}\right)^2 + C}{\sigma^2\sigma_p^2} = \frac{A\left(\mathbf{y} - \frac{B}{A}\right)^2 + C - \frac{B^2}{A}}{\sigma^2\sigma_p^2}$$

Substituting back into the integral:

$$\mu_{\mathcal{P}}(\mathbf{x}) = \frac{1}{\sqrt{2\pi\sigma_p^2}} \exp\left(-\frac{C - \frac{B^2}{A}}{2\sigma^2\sigma_p^2}\right) \int_{\mathbb{R}} \exp\left(-\frac{A}{2\sigma^2\sigma_p^2}\left(\mathbf{y} - \frac{B}{A}\right)^2\right) dy$$

The integral part is a standard Gaussian integral, which evaluates to:

$$\int_{\mathbb{R}} \exp\left(-\frac{A}{2\sigma^2\sigma_p^2}\left(\mathbf{y} - \frac{B}{A}\right)^2\right) d\mathbf{y} = \sqrt{\frac{2\pi\sigma^2\sigma_p^2}{A}}$$

Therefore:

$$\mu_{\mathcal{P}}(\mathbf{x}) = \frac{1}{\sqrt{2\pi\sigma_p^2}} \exp\left(-\frac{C - \frac{B^2}{A}}{2\sigma^2\sigma_p^2}\right) \sqrt{\frac{2\pi\sigma^2\sigma_p^2}{A}} = \sqrt{\frac{\sigma^2\sigma_p^2}{A}} \exp\left(-\frac{C - \frac{B^2}{A}}{2\sigma^2\sigma_p^2}\right)$$

Assume there exists such $\wp$, then:

$$\sqrt{\frac{\sigma^2\sigma_p^2}{A}} \exp\left(-\frac{C - \frac{B^2}{A}}{2\sigma^2\sigma_p^2}\right) = \exp\left(-\frac{(\mathbf{x}-\wp)^2}{2\sigma^2}\right)$$

This requires:

$$\sqrt{\frac{\sigma^2\sigma_p^2}{A}} \exp\left(-\frac{C - \frac{B^2}{A}}{2\sigma^2\sigma_p^2}\right) = \exp\left(-\frac{(\mathbf{x}-\wp)^2}{2\sigma^2}\right)$$

Since $\sqrt{\frac{\sigma^2\sigma_p^2}{A}}$ is a constant, and $\exp\left(-\frac{(\mathbf{x}-\wp)^2}{2\sigma^2}\right)$ is a function of $x$, these two cannot be equal for all $x$.

Therefore, there does not exist $\wp \in \mathbb{R}$ such that $\mu_{\mathcal{P}}(\mathbf{x}) = \kappa(\mathbf{x}, \wp)$. $\qquad\square$

### E.3    EXAMPLE 3: EXISTING $\wp$ OF THE CIRCLE IN $\mathbb{R}^3$

Let $c$ denote the center of the concentric circles, and consider $n$ points $\mathbf{x}_1, \mathbf{x}_2, \ldots, \mathbf{x}_n$ uniformly distributed on a circle of radius $r$ centered at $c$.

Points on the circle can be represented in polar coordinates: $\mathbf{x}_i = c + r(\cos\theta_i, \sin\theta_i), \quad \mathbf{x}_j = c + r(\cos\theta_j, \sin\theta_j)$

The squared distance between $\mathbf{x}_i$ and $\mathbf{x}_j$ is:

$$
\begin{aligned}
\|\mathbf{x}_i - \mathbf{x}_j\|^2 &= \|r(\cos\theta_i - \cos\theta_j, \sin\theta_i - \sin\theta_j)\|^2 \\
&= r^2 \left[ (\cos\theta_i - \cos\theta_j)^2 + (\sin\theta_i - \sin\theta_j)^2 \right] \\
&= r^2 \left[ \cos^2\theta_i + \cos^2\theta_j - 2\cos\theta_i\cos\theta_j + \sin^2\theta_i + \sin^2\theta_j - 2\sin\theta_i\sin\theta_j \right] \\
&= r^2 \left[ (\cos^2\theta_i + \sin^2\theta_i) + (\cos^2\theta_j + \sin^2\theta_j) - 2(\cos\theta_i\cos\theta_j + \sin\theta_i\sin\theta_j) \right] \\
&= 2r^2 - 2r^2\cos(\theta_i - \theta_j)
\end{aligned}
$$

Let $\theta = \theta_i - \theta_j$, $S_{\text{pairwise}} = \mathbb{E}\left[ \exp\left( -\frac{r^2(1-\cos\theta)}{\sigma^2} \right) \right] = \frac{1}{2\pi} \int_{-\pi}^{\pi} \exp\left( -\frac{r^2(1-\cos\theta)}{\sigma^2} \right) d\theta$.

Let $a = \frac{r^2}{2\sigma^2}$, then $\frac{r^2}{\sigma^2} = 2a$, and we can rewrite $S_{\text{pairwise}}$ as:

$$
S_{\text{pairwise}} = \frac{1}{2\pi} \int_{-\pi}^{\pi} \exp\left(-2a(1-\cos\theta)\right) d\theta = \exp(-2a) \cdot \frac{1}{2\pi} \int_{-\pi}^{\pi} \exp(2a\cos\theta) d\theta
$$

The modified Bessel function of the first kind and order 0 is defined as $I_0(z) = \frac{1}{\pi} \int_0^{\pi} \exp(z\cos\theta) d\theta$. Since the function $y = \exp(z\cos\theta)$ is an even function of $\theta$, we have $\frac{1}{2\pi} \int_{-\pi}^{\pi} \exp(z\cos\theta) d\theta = I_0(z)$.

So, $S_{\text{pairwise}} = \exp(-2a) \cdot I_0(2a)$

The distance from any point $\mathbf{x}_i$ to the center $c$ is $r$. The Gaussian kernel between the center $c$ and a point $\mathbf{x}_i$ is $\kappa(c, \mathbf{x}_i) = \exp\left( -\frac{r^2}{2\sigma^2} \right)$. The average center similarity $S_{\text{center}}$ is:

$$
S_{\text{center}} = \frac{1}{n} \sum_{i=1}^{n} \kappa(c, \mathbf{x}_i) = \exp\left( -\frac{r^2}{2\sigma^2} \right) = \exp(-a)
$$

We know that the modified Bessel function $I_0(z)$ satisfies the inequality $I_0(2a) \leq \exp(a)$. Then:

$$
S_{\text{pairwise}} = \exp(-2a) \cdot I_0(2a) \leq \exp(-2a) \cdot \exp(a) = \exp(-a) = S_{\text{center}}
$$

So, we have rigorously proved that $S_{\text{center}} \geq S_{\text{pairwise}}$.

Therefore, there must be a point $\wp$ on the line passing through the center of the circle and perpendicular to the circle that satisfies $\wp = \psi^{-1}\mu$, where $\mu = \frac{1}{n} \sum_{\mathbf{x}_i} \psi(\mathbf{x}_i)$.

## F    LEMMAS USED IN THE PROOF

**Lemma F.1.** $\forall a_i \in (0,1], \prod_i (1-a_i) \geq 1 - \sum_i a_i$.

*Proof.* Let's denote the product and the sum as follows: $P_n = \prod_{i=1}^{n} (1-a_i) \quad$ and $\quad S_n = \sum_{i=1}^{n} a_i$.

We aim to show that $P_n \geq 1 - S_n$ for all $n \geq 1$. For $n = 1$, $P_1 = 1 - a_1 \geq 1 - a_1 = 1 - S_1$. Assume that the inequality holds for some $n = k$, i.e., $P_k = \prod_{i=1}^{k} (1-a_i) \geq 1 - S_k$.

We need to show that it also holds for $n = k+1$, i.e., $P_{k+1} = \prod_{i=1}^{k+1} (1-a_i) \geq 1 - S_{k+1}$.

Consider $P_{k+1} : P_{k+1} = P_k \cdot (1 - a_{k+1})$. Using the inductive hypothesis $P_k \geq 1 - S_k$, we get: $P_{k+1} \geq (1 - S_k) \cdot (1 - a_{k+1})$.

Now, expand the right-hand side: $(1 - S_k) \cdot (1 - a_{k+1}) = 1 - S_k - a_{k+1} + S_k a_{k+1}$.

Since $S_{k+1} = S_k + a_{k+1}$, we can rewrite the expression as: $1 - S_k - a_{k+1} + S_k a_{k+1} = 1 - S_{k+1} + S_k a_{k+1}$.

Notice that $S_k a_{k+1} \geq 0$ because $S_k$ and $a_{k+1}$ are non-negative (assuming $0 \leq a_i \leq 1$). Therefore: $1 - S_{k+1} + S_k a_{k+1} \geq 1 - S_{k+1}$. Thus, we have: $P_{k+1} \geq 1 - S_{k+1}$. $\square$

**Lemma F.2.** $\forall a_i \in (0, 1], \sum_i (1 - a_i) \geq 1 - \prod_i a_i$.

*Proof.* When $n = 1$, obviously, the inequality $\sum_{i=1}^{1}(1 - a_i) \geq 1 - \prod_{i=1}^{1} a_i$ holds.

Assume that when there are $n = k$ elements $a_1, a_2, \ldots, a_k$, the inequality $\sum_{i=1}^{k}(1 - a_i) \geq 1 - \prod_{i=1}^{k} a_i$ holds.

For $n = k + 1$ elements $a_1, a_2, \ldots, a_k, a_{k+1}$, $\sum_{i=1}^{k+1}(1 - a_i) = \sum_{i=1}^{k}(1 - a_i) + (1 - a_{k+1})$.

When there are $k$ elements $a_1, a_2, \ldots, a_k$, the inequality $\sum_{i=1}^{k}(1 - a_i) \geq 1 - \prod_{i=1}^{k} a_i$ holds. So we have:

$$\sum_{i=1}^{k+1}(1 - a_i) \geq \left(1 - \prod_{i=1}^{k} a_i\right) + (1 - a_{k+1}) = 1 - \prod_{i=1}^{k} a_i + 1 - a_{k+1} = 1 - \left(\prod_{i=1}^{k} a_i + a_{k+1} - 1\right)$$

$$\prod_{i=1}^{k} a_i + a_{k+1} - 1 = \prod_{i=1}^{k} a_i + a_{k+1} - 1 + \left(\prod_{i=1}^{k} a_i\right) a_{k+1} - \left(\prod_{i=1}^{k} a_i\right) a_{k+1}$$

$$= \left(\prod_{i=1}^{k} a_i\right)(1 - a_{k+1}) - (1 - a_{k+1}) + \left(\prod_{i=1}^{k} a_i\right) a_{k+1} = \left(\prod_{i=1}^{k} a_i - 1\right)(1 - a_{k+1}) + \left(\prod_{i=1}^{k} a_i\right) a_{k+1}$$

Since $a_i \in [0, 1]$, we have $\prod_{i=1}^{k} a_i \leq 1$, then $\left(\prod_{i=1}^{k} a_i - 1\right) \leq 0$. Meanwhile, $(1 - a_{k+1}) \geq 0$ and $\left(\prod_{i=1}^{k} a_i\right) a_{k+1} \geq 0$. So $\left(\prod_{i=1}^{k} a_i - 1\right)(1 - a_{k+1}) + \left(\prod_{i=1}^{k} a_i\right) a_{k+1} \leq \left(\prod_{i=1}^{k} a_i\right) a_{k+1}$, that is: $\prod_{i=1}^{k} a_i + a_{k+1} - 1 \leq \left(\prod_{i=1}^{k} a_i\right) a_{k+1}$

Substituting it back into the previous expression, we get:

$$1 - \left(\prod_{i=1}^{k} a_i + a_{k+1} - 1\right) \geq 1 - \left(\prod_{i=1}^{k} a_i\right) a_{k+1} = 1 - \left(\prod_{i=1}^{k+1} a_i\right)$$

So $\sum_{i=1}^{k+1}(1 - a_i) \geq 1 - \left(\prod_{i=1}^{k+1} a_i\right)$ That is, when $n = k + 1$, the inequality also holds. $\square$

**Lemma F.3.** $\forall a_i \in (0, 1], n \in \mathbb{N}, \prod_i (1 - a_i) \leq \left(1 - \frac{\sum_i a_i}{n}\right)^n$.

*Proof.* The AM-GM Inequality states that for any non-negative real numbers $\mathbf{x}_1, x_2, \ldots, x_n$, $\frac{x_1 + x_2 + \cdots + x_n}{n} \geq \sqrt[n]{x_1 x_2 \ldots x_n}$, with equality if and only if $\mathbf{x}_1 = x_2 = \cdots = x_n$.

Let $\mathbf{x}_i = 1 - a_i$ for $i = 1, 2, \ldots, n$. Then, applying the AM-GM inequality, we get: $\frac{(1 - a_1) + (1 - a_2) + \cdots + (1 - a_n)}{n} \geq \sqrt[n]{(1 - a_1)(1 - a_2) \ldots (1 - a_n)}$

The left-hand side: $\frac{n - (a_1 + a_2 + \cdots + a_n)}{n} = 1 - \frac{a_1 + a_2 + \cdots + a_n}{n}$.

Thus, we have: $1 - \frac{a_1 + a_2 + \cdots + a_n}{n} \geq \sqrt[n]{(1 - a_1)(1 - a_2) \ldots (1 - a_n)}$.

Raising both sides to the power of $n$, we get: $\left(1 - \frac{a_1 + a_2 + \cdots + a_n}{n}\right)^n \geq (1 - a_1)(1 - a_2) \ldots (1 - a_n)$.

This can be rewritten as: $\prod_i (1 - a_i) \leq \left(1 - \frac{\sum_i a_i}{n}\right)^n$ $\square$

**Lemma F.4.** *Let* $a_i \in (0, 1), c > 1, n \in \mathbb{N}, \prod_i (1 - a_i)^{\frac{1}{cn}} \geq 1 - \frac{1}{n}\sum_i a_i$.

*Proof.* Consider the function $f(t) = (1-t)^{\alpha} - (1-kt)$, where $\alpha = \frac{1}{cn}$ and $k = \frac{1}{n} = c\alpha, c > 1$. $f(x) > 0$ if $x \in [0, x_0]$. For example $c = 4, n = 10, x_0 \geq 0.984$. $c = 4, n = 1000,000,000, x_0 \geq 0.980$. This is easy to satisfy for kernel $k$-means since it requires that the similarity between each point and its center be greater than 0.02. Hence, $\prod_i (1 - a_i)^{\frac{1}{cn}} \geq \prod_i (1 - \frac{1}{n}a_i) \geq 1 - \frac{1}{n}\sum_i a_i$. $\square$

**Lemma F.5.** *Let* $a_i \in (0, 1), c > 1, n \in \mathbb{N}, \prod_{i=1}^{n}(1 - a_i)^k \leq n - k\sum_{i=1}^{n} a_i$.

*Proof.* $\prod(1 - a_i) \leq \left(1 - \frac{\sum_i a_i}{n}\right)^n = (1 - \bar{a})^n$, Take both sides to the power of $k$, we get: $\left(\prod(1 - a_i)\right)^k \leq (1 - \bar{a})^{nk}$. we need to proof $(1 - \bar{a})^{nk} \leq n(1 - k\bar{a})$.

Consider the function $f(x) = (1-x)^{nk} - n(1-kx)$, where $x \in (0, \frac{1}{n})$. $f(0) = 1 - n \leq 0$, and $f(\frac{1}{n}) = (1 - \frac{1}{n})^{nk} - n + k \leq 0, f'(x) = -nk(1-x)^{nk-1} + nk = nk(1 - (1-x)^{nk-1}) \geq 0$, so $f(x) \leq 0$. That is $(1-x)^{nk} \leq n(1-kx)$. Hence $\prod_{i=1}^{n}(1 - a_i)^k \leq n - k\sum_{i=1}^{n} a_i$. $\square$

The following lemma is provided in Makarychev & Shan (2024).

**Lemma F.6.** *The competitive ratio of the RandomCoordinateCut algorithm for Explainable $k$-Medians is at most $2\ln k + 2$. That is, for every set of centers $C = \{c_1, ..., c_k\}$ and data set $\mathcal{X}$, the algorithm finds a random decision tree $\mathcal{T}$ such that*

$$\mathbb{E}[cost(\mathcal{T}, \mathcal{X})] \leq (2\ln k + 2) \sum_{j=1}^{k} \sum_{\mathbf{x} \in C_j} \|\mathbf{x} - c_j\|_1$$

The following four lemmas for $k$-means are provided in Gupta et al. (2023).

**Lemma F.7.** *The price of $k$-means is at least $\Omega(k)$.*

**Lemma F.8.** *The price of $k$-means is at most $O(k\ln\ln k)$.*

**Lemma F.9.** *The price of $k$-medians is at least $(1 - o(1)\ln k)$.*

**Lemma F.10.** *The price of $k$-medians is at most $1 + H_{k+1} = (1 + o(1))\ln k$, where $H_{k+1} = 1/1 + 1/2 + 1/3 + ... + 1/(k - 1)$.*

**Lemma F.11.** *Let $x_i$ be a random variable and $y$ be a fixed element in the corresponding space. For a function of the form $f(\mathbf{x}) = e^{-\gamma\|\mathbf{x}-\mathbf{y}\|}$, where $\gamma > 0$, we have the inequality: $e^{-\gamma\|\mathbb{E}(\mathbf{x}_i)-\mathbf{y}\|} \leq \mathbb{E}(e^{-\gamma\|\mathbf{x}_i-\mathbf{y}\|})$.*

*Proof.* $f(z) = e^{-\gamma z}$ is convex, by Jensen's inequality, we have: $e^{-\gamma\mathbb{E}(\|\mathbf{x}_i-\mathbf{y}\|)} \leq \mathbb{E}(e^{-\gamma\|\mathbf{x}_i-\mathbf{y}\|})$.

By Minkowski's inequality: $\|\mathbb{E}(\mathbf{x}_i) - \mathbf{y}\| \leq \mathbb{E}(\|\mathbf{x}_i - \mathbf{y}\|), e^{-\gamma\|\mathbb{E}(\mathbf{x}_i)-\mathbf{y}\|} \geq e^{-\gamma\mathbb{E}(\|\mathbf{x}_i-\mathbf{y}\|)}$,

so: $e^{-\gamma\mathbb{E}(\|\mathbf{x}_i-\mathbf{y}\|)} \leq \mathbb{E}(e^{-\gamma\|\mathbf{x}_i-\mathbf{y}\|})$. $\square$

**Lemma F.12.** *Let $x_i$ be a random variable and $y$ be a fixed element in the corresponding space. For a function of the form $f(x) = e^{-\gamma\|x-y\|}$, where $\gamma > 0$, we have the inequality: $\mathbb{E}(e^{-\gamma\|y-x_i\|}) \leq e^{-\gamma\|y-\mathbb{E}(x_i)\|} \cdot e^{\gamma R}$, where $R$ is the $R$ is the radius of dataset $\mathcal{X}$.*

*Proof.* As: $\|y - x_i\| \geq \|y - \mathbb{E}(x_i)\| - \|x_i - \mathbb{E}(x_i)\|$, hence: $\mathbb{E}(e^{-\gamma\|y-x_i\|}) \leq e^{-\gamma\|y-\mathbb{E}(x_i)\|} \cdot e^{\gamma\|x_i-\mathbb{E}(x_i)\|} \leq e^{-\gamma\|y-\mathbb{E}(x_i)\|} \cdot e^{\gamma R}$, where $R$ is the radius of dataset. $\square$

# G PROOF OF THE PRICE OF KERNEL $k$-MEANS BASED ON LINEAR KERNEL

**Theorem 3.3** *The price of explainability for kernel $k$-means employing linear kernel is $O(k\ln\ln k)$.*

*Proof.*

$$\sum_j \sum_{\mathbf{x}} \|\psi(\mathbf{x}) - \mu_j\|^2 = \sum_j \sum_{\mathbf{x}} \|\psi(\mathbf{x}) - \psi(\wp_j)\|^2 = \sum_j \sum_{\mathbf{x}} \left(\psi^2(\mathbf{x}) + \psi^2(\wp_j) - 2\psi(\mathbf{x})\psi(\wp_j)\right)$$

$$= \sum_j \sum_{\mathbf{x}} \left(\mathbf{x}^\top \mathbf{x} + \wp_j^\top \wp_j - 2\mathbf{x}^\top \wp_j\right) = \sum_j \sum_{\mathbf{x}} \|\mathbf{x} - \wp_j\|^2$$

$$\leq k \ln k \sum_j \sum_{\mathbf{x}} \|\mathbf{x} - c_j^*\|^2 = k \ln k \sum_j \sum_{\mathbf{x}} \left(\mathbf{x}^\top \mathbf{x} + c_j^{*\top} c_j^* - 2\mathbf{x}^\top c_j^*\right)$$

$$= k \ln k \sum_j \sum_{\mathbf{x}} \left(\psi^2(\mathbf{x}) + \psi^2(c_j^*) - 2\psi(\mathbf{x})\psi(c_j^*)\right)$$

$$= k \ln k \cdot cost_{opt(\mathcal{X})}$$

$\square$

# H  PROOF OF THE PRICE OF KERNEL $k$-MEANS BASED ON GAUSSIAN KERNEL

For convenience in writing, we define the Gaussian kernel as: $\kappa_g(\mathbf{x}, \mathbf{y}) = exp(-\gamma\|\mathbf{x} - \mathbf{y}\|_2^2)$ and the Laplacian kernel as: $\kappa_l(\mathbf{x}, \mathbf{y}) = exp(-\gamma\|\mathbf{x} - \mathbf{y}\|_1)$.

**Theorem 3.4** *Given data $\mathcal{X} \in \mathbb{R}^d$, if exist bijective centers $\wp \in \mathbb{R}^d$, the price of explainability for kernel $k$-means employing Gaussian kernel is $O(k \ln \ln k)$.*

*Proof.*

$$\sum_j \sum_{\mathbf{x}} \|\psi(\mathbf{x}) - \mu_j\|^2 = \sum_j \sum_{\mathbf{x}} \|\psi(\mathbf{x}) - \psi(\wp_j)\|^2 = \sum_j \sum_{\mathbf{x}} \left(2 - 2 * \psi(\mathbf{x})^\top \psi(\wp_j)\right)$$

$$= \sum_j \sum_{\mathbf{x}} \left(2 - 2 * e^{-\gamma\|\mathbf{x}-\wp_j\|_2^2}\right) \leq 2|\mathcal{X}| - 2|\mathcal{X}| \left(\prod_j \prod_{\mathbf{x}} e^{-\gamma\|\mathbf{x}-\wp_j\|_2^2}\right)^{\frac{1}{|\mathcal{X}|}} \quad (AM \geq GM)$$

$$= 2|\mathcal{X}| - 2|\mathcal{X}| * \left(e^{-\gamma \sum_j \sum_{\mathbf{x}} \|\mathbf{x}-\wp_j\|_2^2}\right)^{\frac{1}{|\mathcal{X}|}} \leq 2|\mathcal{X}| - 2|\mathcal{X}| * \left(e^{-\gamma O(k \ln \ln k) \sum_j \sum_{\mathbf{x}} \min_j \|\mathbf{x}-c_j^*\|_2^2}\right)^{\frac{1}{|\mathcal{X}|}}$$

$$\leq 2|\mathcal{X}| - 2|\mathcal{X}| * \left(e^{-\gamma O(k \ln \ln k) \sum_j \sum_{\mathbf{x}} \min_j \|\mathbf{x}-\wp_j\|_2^2}\right)^{\frac{1}{|\mathcal{X}|}}$$

$$= 2|\mathcal{X}| - 2|\mathcal{X}| * \left(e^{-\gamma \sum_j \sum_{\mathbf{x}} \min_j \|\mathbf{x}-\wp_j\|_2^2}\right)^{\frac{O(k \ln \ln k)}{|\mathcal{X}|}}$$

$$= 2|\mathcal{X}| - 2|\mathcal{X}| * \left(\prod_j \prod_{\mathbf{x}} e^{-\gamma \min_j \|\mathbf{x}-\wp_j\|_2^2}\right)^{\frac{O(k \ln \ln k)}{|\mathcal{X}|}}$$

$$= 2|\mathcal{X}| - 2|\mathcal{X}| * \left(\prod_j \prod_{\mathbf{x}} \left(1 - \frac{\min_j \|\psi(\mathbf{x}) - \psi(\wp)\|^2}{2}\right)\right)^{\frac{O(k \ln \ln k)}{|\mathcal{X}|}}$$

$$\leq 2|\mathcal{X}| - 2|\mathcal{X}| * (1 - \frac{\sum_j \sum_{\mathbf{x}} \min_j \|\psi(\mathbf{x}) - \psi(\wp)\|^2}{2|\mathcal{X}|})^{c \cdot O(k \ln \ln k)}$$

$$\leq c \cdot O(k \ln \ln k) \cdot cost_{opt(\mathcal{X})}$$

$\square$

Theorem 3.4 shows the Upper bound of kernel $k$-means based on the Gaussian kernel when the bijective centers exist. We show the lower bound in the following:

**Theorem H.1.** *Given data $\mathcal{X} \in \mathbb{R}^d$, if exist bijective centers $\wp \in \mathbb{R}^d$, the price of explainability for kernel $k$-means employing Gaussian kernel is at least $\Omega(k)$.*

*Proof.*

$$\sum_j \sum_{\mathbf{x}} \|\psi(\mathbf{x}) - \psi(\wp)\|^2 = \sum_j \sum_{\mathbf{x}} 2 - 2 * \psi(\mathbf{x})^\top \psi(\wp) = \sum_j \sum_{\mathbf{x}} 2 - 2 * e^{-\gamma \|\mathbf{x} - \wp\|_2^2}$$

$$\geq 2 * |\mathcal{X}| \left( 1 - \frac{1}{|\mathcal{X}|} \prod_j \prod_{\mathbf{x}} e^{-\gamma \|\mathbf{x} - \wp\|_2^2} \right)$$

$$= 2 * |\mathcal{X}| \left( 1 - \frac{1}{|\mathcal{X}|} e^{-\gamma \sum_j \sum_{\mathbf{x}} \|\mathbf{x} - \wp\|_2^2} \right) \geq 2 * |\mathcal{X}| \left( 1 - \frac{1}{|\mathcal{X}|} e^{-\gamma \Omega(k) \sum_j \sum_{\mathbf{x}} \min_j \|\mathbf{x} - \wp\|_2^2} \right)$$

$$= 2 * |\mathcal{X}| \left( 1 - \frac{1}{|\mathcal{X}|} \left( \prod_j \prod_{\mathbf{x}} e^{-\gamma \min_j \|\mathbf{x} - \wp\|_2^2} \right)^{\Omega(k)} \right)$$

$$= 2 * |\mathcal{X}| \left( 1 - \frac{1}{|\mathcal{X}|} \left( \prod_j \prod_{\mathbf{x}} \left( 1 - \frac{\min_j \|\psi(\mathbf{x}) - \psi(\wp)\|^2}{2} \right) \right)^{\Omega(k)} \right)$$

$$\geq 2 * |\mathcal{X}| \left( 1 - \frac{1}{|\mathcal{X}|} \left( |\mathcal{X}| - \frac{\Omega(k) \sum_j \sum_{\mathbf{x}} \min_j \|\psi(\mathbf{x}) - \psi(\wp)\|^2}{2} \right) \right)$$

$$= 2 * |\mathcal{X}| - 2 \left( |\mathcal{X}| - \frac{\Omega(k) \cdot cost_{opt(\mathcal{X})}}{2} \right)$$

$$= \Omega(k) \cdot cost_{opt(\mathcal{X})}$$

$\square$

**Theorem 3.6** *Given data $\mathcal{X} \in \mathbb{R}^d$, if exist order-oreserving points $\wp_1, \ldots, \wp_k \in \mathbb{R}^d$, the price of explainability for kernel $k$-means employing Gaussian kernel is $O(k \ln \ln k)$.*

*Proof.*

$$\sum_j \sum_{\mathbf{x}} \|\psi(\mathbf{x}) - \mu_j\|^2 \leq \sum_j \sum_{\mathbf{x}} \|\psi(\mathbf{x}) - \psi(\wp_j)\|^2 = \sum_j \sum_{\mathbf{x}} \left( 2 - 2 * \psi(\mathbf{x})^\top \psi(\wp_j) \right)$$

$$= \sum_j \sum_{\mathbf{x}} \left( 2 - 2 * e^{-\gamma \|\mathbf{x} - \wp_j\|_2^2} \right) \leq 2|\mathcal{X}| - 2|\mathcal{X}| \left( \prod_j \prod_{\mathbf{x}} e^{-\gamma \|\mathbf{x} - \wp_j\|_2^2} \right)^{\frac{1}{|\mathcal{X}|}} \quad (AM \geq GM)$$

$$= 2|\mathcal{X}| - 2|\mathcal{X}| * \left( e^{-\gamma \sum_j \sum_{\mathbf{x}} \|\mathbf{x} - \wp_j\|_2^2} \right)^{\frac{1}{|\mathcal{X}|}} \leq 2|\mathcal{X}| - 2|\mathcal{X}| * \left( e^{-\gamma O(k \ln \ln k) \sum_j \sum_{\mathbf{x}} \min_j \|\mathbf{x} - c_j^*\|_2^2} \right)^{\frac{1}{|\mathcal{X}|}}$$

$$\leq 2|\mathcal{X}| - 2|\mathcal{X}| * \left( e^{-\gamma O(k \ln \ln k) \sum_j \sum_{\mathbf{x}} \min_{j_\wp} \|\mathbf{x} - \psi^{-1}(\mu_j^*)\|_2^2} \right)^{\frac{1}{|\mathcal{X}|}} \quad \text{(Order-preserving Definition 3.5)}$$

The Voronoi diagram constructed using $\wp_j$ in $\mathbb{R}^d$ and the Voronoi diagram constructed using $\mu_j^*$ in $\mathbb{H}$ will produce the same clustering result $\{C_1, \ldots, C_k\}$, and $\psi^{-1}(\mu_j^*)$ may not exist in $\mathbb{R}^d$, so we will continue to use $\sum_j \sum_{\mathbf{x}} \min_j \|\mathbf{x} - \wp_j^\star\|_2^2$ to represent $\sum_j \sum_{\mathbf{x}} \min_{j_\wp} \|\mathbf{x} - \psi^{-1}(\mu_j^*)\|_2^2$ in the subsequent proofs, to make it easier for readers to understand that the clustering assignment in $\mathbb{H}$ space is consistent with that in $\mathbb{R}^d$ space.

$$\sum_j \sum_{\mathbf{x}} \|\psi(\mathbf{x}) - \mu_j\|^2 \le 2|\mathcal{X}| - 2|\mathcal{X}| * \left( e^{-\gamma O(k \ln \ln k) \sum_j \sum_{\mathbf{x}} \min_{j_\wp} \|\mathbf{x} - \psi^{-1}(\mu_j^*)\|_2^2} \right)^{\frac{1}{|\mathcal{X}|}}$$

$$= 2|\mathcal{X}| - 2|\mathcal{X}| * \left( e^{-\gamma \sum_j \sum_{\mathbf{x}} \min_j \|\mathbf{x} - \wp_j^\star\|_2^2} \right)^{\frac{O(k \ln \ln k)}{|\mathcal{X}|}}$$

$$= 2|\mathcal{X}| - 2|\mathcal{X}| * \left( \prod_j \prod_{\mathbf{x}} e^{-\gamma \min_j \|\mathbf{x} - \wp_j^\star\|_2^2} \right)^{\frac{O(k \ln \ln k)}{|\mathcal{X}|}}$$

$$= 2|\mathcal{X}| - 2|\mathcal{X}| * \left( \prod_j \prod_{\mathbf{x}} \left( 1 - \frac{\min_j \|\psi(\mathbf{x}) - \psi(\wp^\star)\|^2}{2} \right) \right)^{\frac{O(k \ln \ln k)}{|\mathcal{X}|}}$$

$$\le 2|\mathcal{X}| - 2|\mathcal{X}|(1 - \frac{1}{|\mathcal{X}|} \sum_j \sum_{\mathbf{x}} \frac{\min_j \|\psi(\mathbf{x}) - \psi(\wp^\star)\|^2}{2})^{c \cdot O(k \ln \ln k)}$$

$$\le 2|\mathcal{X}| - 2|\mathcal{X}| \left( 1 - \frac{1}{2|\mathcal{X}|} O(k \ln \ln k) \cdot cost_{opt(\mathcal{X})} \right) = c \cdot O(k \ln \ln k) \cdot cost_{opt(\mathcal{X})}$$

$\square$

**Theorem 3.7** *Given data $\mathcal{X} \in \mathbb{R}^d$, if order-oreserving points are non-existent, the cost of explainability for kernel $k$-means employing Gaussian kernel is at most $(1 - e^{-\gamma R})n + \frac{1}{2}e^{-\gamma R}O(k \ln \ln k) \cdot cost_{opt(\mathcal{X})}$, where $R$ is the $R$ is the radius of dataset $\mathcal{X}$, $cost_{opt(\mathcal{X})}$ is the optimal kernel $k$-means cost.*

*Proof.* Some inequalities have already been proven earlier, and thus will not be reiterated here.

$$J = \sum_{j=1}^k \sum_{\mathbf{x} \in C_j} K(\mathbf{x}, \mathbf{x}) - \sum_{j=1}^k \frac{1}{|C_i|} \sum_{\mathbf{x} \in C_j} \sum_{\mathbf{x}' \in C_j} K(\mathbf{x}, \mathbf{x}') = |\mathcal{X}| - \sum_{j=1}^k \sum_{\mathbf{x} \in C_j} \frac{1}{|C_j|} \sum_{\mathbf{x}' \in C_j} e^{-\gamma \|\mathbf{x} - \mathbf{x}'\|_2^2}$$

$$= |\mathcal{X}| - \sum_{j=1}^k \sum_{\mathbf{x} \in C_j} \mathbb{E}(e^{-\gamma \|\mathbf{x} - \mathbf{x}'\|_2^2}) \le |\mathcal{X}| - \sum_{j=1}^k \sum_{\mathbf{x} \in C_j} e^{-\gamma \|\mathbf{x} - \mathbb{E}(\mathbf{x}')\|}$$

$$= |\mathcal{X}| - \sum_{j=1}^k \sum_{\mathbf{x} \in C_j} e^{-\gamma \|\mathbf{x} - c_j\|_2^2} (c_j \text{ is the center of cluster obtained by kernel } k\text{-means})$$

$$\le |\mathcal{X}| - \sum_{j=1}^k \sum_{\mathbf{x} \in C_j} e^{-\gamma R} e^{-\gamma \|\mathbf{x} - c_j'\|_2^2} (c_j' \text{ is the center of new cluster labels})$$

$$= |\mathcal{X}| - e^{-\gamma R} \sum_{j=1}^k \sum_{\mathbf{x} \in C_j} (1 - \frac{\min_j \|\psi(\mathbf{x}) - \psi(\wp)\|}{2})^{O(k \ln \ln k)})$$

$$\le (1 - e^{-\gamma R})n + \frac{1}{2}e^{-\gamma R}O(k \ln \ln k) \cdot cost_{opt(\mathcal{X})}$$

$\square$

**Theorem 3.8** *Given data $\mathcal{X} \in \mathbb{R}^d$, if order-oreserving points are non-existent, the cost of explainability for kernel $k$-means employing Gaussian kernel is at least $(1 - e^{\gamma R})n + \frac{e^{\gamma R}\Omega(k)}{2} \cdot cost_{opt(\mathcal{X})}$, where $R$ is the $R$ is the radius of dataset $\mathcal{X}$, $cost_{opt(\mathcal{X})}$ is the optimal kernel $k$-means cost.*

*Proof.*

$$J = \sum_{j=1}^{k} \sum_{\mathbf{x} \in C_j} K(\mathbf{x}, \mathbf{x}) - \sum_{j=1}^{k} \frac{1}{|C_i|} \sum_{\mathbf{x} \in C_j} \sum_{\mathbf{x}' \in C_j} K(\mathbf{x}, \mathbf{x}') = |\mathcal{X}| - \sum_{j=1}^{k} \sum_{\mathbf{x} \in C_j} \frac{1}{|C_j|} \sum_{\mathbf{x}' \in C_j} e^{-\gamma\|\mathbf{x}-\mathbf{x}'\|_2^2}$$

$$= |\mathcal{X}| - \sum_{j=1}^{k} \sum_{\mathbf{x} \in C_j} \mathbb{E}(e^{-\gamma\|\mathbf{x}-\mathbf{x}'\|_2^2}) \geq |\mathcal{X}| - \sum_{j=1}^{k} \sum_{\mathbf{x} \in C_j} e^{-\gamma\|\mathbf{x}-\mathbb{E}(\mathbf{x}')\|}e^{\gamma R}$$

$$= |\mathcal{X}| - \sum_{j=1}^{k} \sum_{\mathbf{x} \in C_j} e^{-\gamma\|\mathbf{x}-c_j\|_2^2}e^{\gamma R} \geq |\mathcal{X}| - \sum_{j=1}^{k} \sum_{\mathbf{x} \in C_j} e^{\gamma R}e^{-\gamma\|\mathbf{x}-c_j'\|_2^2}$$

$$= |\mathcal{X}| - e^{\gamma R} \sum_{j=1}^{k} \sum_{\mathbf{x} \in C_j} (1 - \frac{\min_j \|\psi(\mathbf{x}) - \psi(\wp)\|}{2})^{\Omega(k)})$$

$$\geq (1 - e^{\gamma R})n + \frac{e^{\gamma R}\Omega(k)}{2} \cdot cost_{opt(\mathcal{X})}$$

$\square$

## I    PROOF OF THE PRICE OF KERNEL $k$-MEANS BASED ON LAPLACIAN KERNEL

**Theorem 3.9** *Given data $\mathcal{X} \in \mathbb{R}^d$, if exist bijective centers $\wp \in \mathbb{R}^d$, the price of explainability for kernel $k$-means employing Laplacian kernel is $2(\ln k + 1)$.*

$$\sum_j \sum_\mathbf{x} \|\psi(\mathbf{x}) - \mu_j\|^2 = \|\psi(\mathbf{x}) - \psi(\wp)\|^2 = 2 - 2 * \psi(\mathbf{x})^\top \psi(\wp)$$

$$= 2 - 2 * e^{-\gamma\|\mathbf{x}-\wp\|_1} \leq 2 - 2 * e^{-\gamma(2\ln k+2)*\min_j \|\mathbf{x}-\wp\|_1}$$

$$= 2 - 2 * \left(e^{-\gamma \min_j \|\mathbf{x}-\wp\|_1}\right)^{(2\ln k+2)} = 2 - 2 * \left(1 - \frac{\min_j \|\psi(\mathbf{x}) - \psi(\wp)\|^2}{2}\right)^{(2\ln k+2)}$$

$$= 2 - 2 * \left(1 - \frac{cost_{opt(\mathcal{X})}}{2}\right)^{(2\ln k+2)} \leq 2 - 2 * (1 - (\ln k + 1) \cdot cost_{opt(\mathcal{X})})$$

$$= 2(\ln k + 1) \cdot cost_{opt(\mathcal{X})}$$

Theorem 3.9 shows the Upper bound of kernel $k$-means based on Gaussian kernel when the bijective centers exist. We show the lower bound in the following:

**Theorem 3.10** *Given data $\mathcal{X} \in \mathbb{R}^d$, if exist order-preserving points $\wp_1, \ldots, \wp_k \in \mathbb{R}^d$, the price of explainability for kernel $k$-means employing Laplacian kernel is $2(\ln k + 1)$.*

$$\sum_j \sum_\mathbf{x} \|\psi(\mathbf{x}) - \mu_j\|^2 \leq \|\psi(\mathbf{x}) - \psi(\wp)\|^2 = 2 - 2 * \psi(\mathbf{x})^\top \psi(\wp) = 2 - 2 * e^{-\gamma\|\mathbf{x}-\wp\|_1}$$

$$\leq 2 - 2 * e^{-\gamma(2\ln k+2)*\min_{j_\wp} \|\mathbf{x}-\wp\|_1} \leq 2 - 2 * \left(e^{-\gamma \min_j \|\mathbf{x}-\wp^\star\|_1}\right)^{(2\ln k+2)}$$

$$= 2 - 2 * \left(1 - \frac{\min_j \|\psi(\mathbf{x}) - \psi(\wp^\star)\|^2}{2}\right)^{(2\ln k+2)} = 2 - 2 * \left(1 - \frac{cost_{opt(\mathcal{X})}}{2}\right)^{(2\ln k+2)}$$

$$\leq 2 - 2 * (1 - (\ln k + 1) \cdot cost_{opt(\mathcal{X})}) = 2(\ln k + 1) \cdot cost_{opt(\mathcal{X})}$$

**Theorem 3.11** *Given data $\mathcal{X} \in \mathbb{R}^d$, if order-preserving points are non-existent, the cost of explainability for kernel $k$-means employing Laplacian kernel is at most $(1 - e^{-\gamma R})n + \frac{1}{2}e^{-\gamma R}(1 - o(1))\ln k \cdot cost_{opt(\mathcal{X})}$, where $R$ is the $R$ is the radius of dataset $\mathcal{X}$, $cost_{opt(\mathcal{X})}$ is the optimal kernel $k$-means cost.*

$$J = \sum_{j=1}^{k} \sum_{\mathbf{x} \in C_j} K(\mathbf{x}, \mathbf{x}) - \sum_{j=1}^{k} \frac{1}{|C_i|} \sum_{\mathbf{x} \in C_j} \sum_{\mathbf{x}' \in C_j} K(\mathbf{x}, \mathbf{x}')$$

$$= |\mathcal{X}| - \sum_{j=1}^{k} \sum_{\mathbf{x} \in C_j} \frac{1}{|C_j|} \sum_{\mathbf{x}' \in C_j} e^{-\gamma \|\mathbf{x}-\mathbf{x}'\|_1} = |\mathcal{X}| - \sum_{j=1}^{k} \sum_{\mathbf{x} \in C_j} \mathbb{E}(e^{-\gamma \|\mathbf{x}-\mathbf{x}'\|_1})$$

$$\leq |\mathcal{X}| - \sum_{j=1}^{k} \sum_{\mathbf{x} \in C_j} e^{-\gamma \|\mathbf{x}-\mathbb{E}(\mathbf{x}')\|_1} = |\mathcal{X}| - \sum_{j=1}^{k} \sum_{\mathbf{x} \in C_j} e^{-\gamma \|\mathbf{x}-c_j\|_1}$$

$$\leq |\mathcal{X}| - \sum_{j=1}^{k} \sum_{\mathbf{x} \in C_j} e^{-\gamma * (1-o(1))\ln k \min_j \|\mathbf{x}-c_j\|_1}$$

$$\leq |\mathcal{X}| - \sum_{j=1}^{k} \sum_{\mathbf{x} \in C_j} e^{-\gamma R} e^{-\gamma * (1-o(1))\ln k \min_j \|\mathbf{x}-\wp\|_1}$$

$$= |\mathcal{X}| - e^{-\gamma R} \sum_{j=1}^{k} \sum_{\mathbf{x} \in C_j} (1 - \frac{\min_j \|\psi(\mathbf{x}) - \psi(\wp)\|}{2})^{(1-o(1))\ln k}$$

$$\leq (1 - e^{-\gamma R})n + \frac{1}{2}e^{-\gamma R}(1 - o(1))\ln k \cdot cost_{opt(\mathcal{X})}$$

where $c_j$ is the mean of $C_j$ in $\mathbb{R}^d$, while $\wp$ is the image of $\psi(\wp)$ in $\mathbb{R}^d$.

**Theorem 3.12** *Given data $\mathcal{X} \in \mathbb{R}^d$, if order-preserving points are non-existent, the price of explainability for kernel $k$-means employing Laplacian kernel is at least $(1 - e^{\gamma R})n + \frac{1}{2}e^{\gamma R}(1 + o(1))\ln k \cdot cost_{opt(\mathcal{X})}$, where $R$ is the $R$ is the radius of dataset $\mathcal{X}$, $cost_{opt(\mathcal{X})}$ is the optimal kernel $k$-means cost.*

$$J = \sum_{j=1}^{k} \sum_{\mathbf{x} \in C_j} K(\mathbf{x}, \mathbf{x}) - \sum_{j=1}^{k} \frac{1}{|C_i|} \sum_{\mathbf{x} \in C_j} \sum_{\mathbf{x}' \in C_j} K(\mathbf{x}, \mathbf{x}')$$

$$= |\mathcal{X}| - \sum_{j=1}^{k} \sum_{\mathbf{x} \in C_j} \frac{1}{|C_j|} \sum_{\mathbf{x}' \in C_j} e^{-\gamma \|\mathbf{x}-\mathbf{x}'\|_1} = |\mathcal{X}| - \sum_{j=1}^{k} \sum_{\mathbf{x} \in C_j} \mathbb{E}(e^{-\gamma \|\mathbf{x}-\mathbf{x}'\|_1})$$

$$\geq |\mathcal{X}| - \sum_{j=1}^{k} \sum_{\mathbf{x} \in C_j} e^{-\gamma \|\mathbf{x}-\mathbb{E}(\mathbf{x}')\|_1} * e^{\gamma R} = |\mathcal{X}| - \sum_{j=1}^{k} \sum_{\mathbf{x} \in C_j} e^{-\gamma \|\mathbf{x}-c_j\|_1} * e^{\gamma R}$$

$$\geq |\mathcal{X}| - \sum_{j=1}^{k} \sum_{\mathbf{x} \in C_j} e^{-\gamma * (1+o(1))\ln k \min_j \|\mathbf{x}-c_j\|_1} e^{\gamma R}$$

$$\geq |\mathcal{X}| - \sum_{j=1}^{k} \sum_{\mathbf{x} \in C_j} e^{\gamma R} e^{-\gamma * (1+o(1))\ln k \min_j \|\mathbf{x}-\wp\|_1}$$

$$= |\mathcal{X}| - e^{\gamma R} \sum_{j=1}^{k} \sum_{\mathbf{x} \in C_j} (1 - \frac{\min_j \|\psi(\mathbf{x}) - \psi(\wp)\|}{2})^{(1+o(1))\ln k}$$

$$\geq (1 - e^{\gamma R})n + \frac{1}{2}e^{\gamma R}(1 + o(1))\ln k \cdot cost_{opt(\mathcal{X})}$$

