# OpenReview forum: "The Price of Explainability for Kernel $k$-means"
_ICLR.cc/2026/Conference — Submitted to ICLR 2026_

### Official Review · Reviewer_tfcv · 2025-10-25

**Soundness:** 3
**Presentation:** 3
**Contribution:** 3
**Rating:** 6
**Confidence:** 4

**Summary:**

This paper investigates the Price of Explainability for kernel k-means clustering. It systematically analyzes the trade-off between explainability and clustering performance across different kernel functions, with a particular focus on the Gaussian kernel and Laplacian kernel. The authors propose a novel "Dual-Threshold Tree (DT²)" algorithm that enables the direct construction of interpretable clustering models in the original space without kernel approximation. Theoretical and experimental results demonstrate that kernel k-means based on the Laplacian kernel incurs a lower explainability cost, and the DT² algorithm outperforms the existing KIMM method on multiple datasets. The study further reveals the intrinsic connections between the Laplacian kernel and k-medians, as well as between the Gaussian kernel and k-means, thereby providing a new perspective for interpretable clustering.

**Strengths:**

1. Strong Innovation: It is the first study to systematically analyze the "Price of Explainability" of different kernel functions from both theoretical and experimental perspectives, and proposes the DT² algorithm to reduce interpretability loss.

2. Tight Integration of Theory and Experiment: The paper verifies findings through rigorous theorem derivation and experimental evidence, achieving high consistency between theoretical results and experimental outcomes.

3. High Method Practicality: The DT² algorithm eliminates the need for kernel approximation, enabling direct construction of interpretable models in the original space and demonstrating good scalability.

4. Novel Research Perspective: It reveals the connections between the Laplacian kernel and k-medians, as well as between the Gaussian kernel and k-means, expanding the theoretical framework for clustering interpretability.

**Weaknesses:**

1. Insufficient Discussion on Limitations: It only analyzes three types of kernels (linear, Gaussian, and Laplacian) and fails to conduct an in-depth exploration of other kernel functions (e.g., neural tangent kernel).

2. Overly Strong Theoretical Assumptions: Some theorems rely on ideal conditions, such as "the existence of an invertible center or order-preserving points," which may be difficult to satisfy in real-world data.

3. Limited Experimental Scale: The verification is only carried out on five small-to-medium-sized datasets, lacking tests in large-scale and high-dimensional scenarios.

**Questions:**

1. Experiments only selected five small-scale datasets. Have you tested high-dimensional or real-scenario data (such as images or text), and do the results remain consistent?

2. The DT² algorithm constructs a dual-threshold tree in the original space to avoid kernel approximation. How does its time complexity compare to that of the KIMM method?

3. The theorem assumes the "existence of order-preserving points". Is this condition verifiable or approximately satisfiable in the distribution of actual high-dimensional data?

---

> ### Author Response · Authors · 2025-11-19
> **rebuttal**
>
> Thanks for your review and suggestions.
>
> A1. Since kernel $k$-means has a time complexity of $O(n^2d)$, it is difficult to scale to large-scale datasets. Therefore, we used the same dataset as existing papers (ICLR 2025). Because the Gaussian kernel and Laplacian kernel are the most commonly used kernel functions, we focused on these two, as do existing papers.
>
> A2. Both have a time complexity of $O(n^2)$. DT$^2$ does not speed up the time; it simply avoids estimating the kernel function, thus offering better interpretability.
>
> A3. This is often easily satisfied unless the data distribution is nested, as shown in Figure 4. High-dimensional data, such as MNIST observed using dimensionality reduction tools like UMAP and t-SNE, do not exhibit this nested structure.

---

### Official Review · Reviewer_ThGX · 2025-10-28

**Soundness:** 1
**Presentation:** 2
**Contribution:** 1
**Rating:** 2
**Confidence:** 3

**Summary:**

The paper studies approximation guarantees for decision trees in clustering with kernel $k$-means. The guarantees are given in terms of the ratio between the clustering cost of a tree, and the optimal cost. This ratio is referred to as the price of explainability (POE). The paper begins by stating the POE for the linear kernel, which reduces the problem to $k$-means. Then, the paper introduces assumptions on the data distribution to derive bounds on the POE. Firstly, it is assumed that the cluster mean in the RKHS has a pre-image in the input space. Secondly, the authors introduce the notion of order-preserving points. For both, the POE is claimed to be $\mathcal{O}(k \ln \ln k)$ for the Gaussian kernel. Thirdly, if such points do not exist, the cost is claimed to depend on and the radius of the dataset. According results are derived for the Laplace kernel. Finally, the paper proposes a greedy cost-minimizing algorithm and evaluates it on a few datasets.

**Strengths:**

Studying kernel clustering and its interpretability under assumptions on the distribution is interesting, as is the notion of order-preserving points.

**Weaknesses:**

The paper lacks clarity and I have concerns regarding the theory.

(1) The notion of an interpretable threshold tree is not formally defined. The paper probably takes the same notion of axis-aligned trees as in prior works, but this should be formalized.

(2) For the Gaussian or Laplace kernel, there can never exist a pre-image for cluster means unless all points in a cluster are identical.
Proof: Take any
$(x_i)_{ i=1,\ldots,n}$.

Then, with $\psi$ denoting the feature map, the mean in RKHS is $\mu = \frac{1}{n} \sum_{i=1}^n \psi(x_i)$. Therefore, $\|\mu\|^2 = \frac{1}{n^2} \sum_{i,j=1}^n \kappa(x_i, x_j) < 1$. However, for any $\rho$ in the input domain, $\| \psi(\rho)\| = \kappa(\rho,\rho) = 1$. Unfortunately, this makes a relevant part of the theory void.

(3) For the case of order-preserving points, the proof of Theorem 3.6 is confusing (Appendix H). Essentially, the proof starts with $(I) = \sum_j \sum_x \| \psi(x) - \mu_j \|^2$, and upper bounds it by $(II) = c \cdot \mathcal{O}(k \ln \ln k) \cdot cost_{opt}$. Unless I am missing something, $(I)$ is already the cost of clustering w.r.t. the means in RKHS, not the cost of clustering with a tree. It is not apparent where in the proof a tree is analyzed. Furthermore, the term $\mathcal{O}(k \ln \ln k)$ appears in several derivations without justification.

(4) The paper also mentions the cost of explainability. How does it give us insight into the price, as alluded to after Theorems 3.7 and 3.8?

(5) It is not clear how order-preserving points would be found in practice, or what are natural conditions for them to exist.

**Questions:**

See questions in weaknesses

---

> ### Author Response · Authors · 2025-11-19
> **rebuttal**
>
> Thanks for your review and suggestions.
>
> # Weaknesses
>
> A1. Thanks for your valuable suggestion.
>
> A2.  There is no point whose feature map is $\mu$, but there exists a point $z$ such that $\forall x, k(x, z) = k(x, \mu)$. This is equivalent when calculating the cost, so $z$ is analyzed as Bijective Centers in the same way.
>
> A3. The $\sum\limits_j\sum\limits_\mathbf{x} \|\psi(\mathbf{x})-\mu_j\|^2$ is the cost of kernel $k$-means. The $\sum\limits_j\sum\limits_\mathbf{x} \|\psi(\mathbf{x})-\mu_j\|^2\leq\sum\limits_j\sum\limits_\mathbf{x} \|\psi(\mathbf{x})-\psi(\wp_j)\|^2$ is valid because $mu_j$ is the optimal value. Term $k\ln\ln k$ is a conclusion from previous work, given in Lemma F.8.
>
> A4. These two are consistent; the price of explainability is the ratio of the cost of explainability to the cost_opt(X).
>
> A5. We will provide our code to find order-preserving points, as we used in the results in Figure 3.

---

> > ### Comment · Reviewer_ThGX · 2025-11-25
> >
> > Thanks for the response. Your answers do not resolve my concerns regarding the theory. In particular, I do not see how a tree is analyzed in the proof of Theorem 3.6. Moreover, in light of the counterexample provided in the review, it seems the authors now propose to switch to a different notion of bijective centers. This needs to be made clearer, with the definitions etc. adjusted accordingly. Regarding cost and price of explainability, my point is this: Without a bound on the optimal cost, a bound on the cost of the explainable clustering cannot be used to bound the price of explainability. This makes it difficult to interpret the result claimed by the authors. I keep my score.

---

### Official Review · Reviewer_iUgn · 2025-10-28

**Soundness:** 2
**Presentation:** 2
**Contribution:** 2
**Rating:** 2
**Confidence:** 4

**Summary:**

This paper investigates the price of explainability in kernel k-means clustering. The authors theoretically analyze explainability costs under linear, Gaussian, and Laplacian kernels with and without bijective or order-preserving points. Furthermore, the paper introduces DT2 ,  avoiding kernel approximations used in KIMM.

**Strengths:**

1. The paper introduces two concepts, Bijective Center and Order-Preserving Points, in the explainability analysis of kernel k-means, which are used to examine the mapping relationship between the RKHS and the original feature space.
2. The proposed Dual-Threshold Tree (DT2) algorithm is a direct improvement over KIMM.

**Weaknesses:**

1. The paper would benefit from a clearer articulation of its research motivation. Clarifying whether the primary aim is to enhance the performance or the explainability of kernel k-means would help readers better appreciate the contribution and positioning of the work.
2. The logical flow of the writing could be further refined. Introducing the concept of kernel k-means explainability before discussing the explainability cost, and explicitly connecting this analysis to the proposed Dual-Threshold Tree algorithm, would make the overall narrative more coherent and accessible.
3. The discussion on explainability itself could be further deepened. While the paper focuses on the price of explainability, it provides limited theoretical insight into what constitutes a “good” or “faithful” explanation. A more explicit exploration of how the proposed method enhances or measures explainability would be highly valuable and is something the community would look forward to seeing.
4. While the current experimental results show modest improvements, especially in Figure 5, the study presents a promising foundation. A deeper investigation into scenarios or datasets where the Dual-Threshold Tree can achieve more pronounced gains would strengthen the empirical impact of the paper.
5. The experimental evaluation could be expanded to better illustrate the interpretability advantages of the proposed method. Including visualizations of explanation results or quantitative comparisons (e.g., Rand index against ground truth) would enrich the assessment of explanation quality.
6. The contribution could be further broadened. Although the paper builds thoughtfully on “Explaining Kernel Clustering via Decision Trees,” extending the theoretical analysis beyond Gaussian and Laplacian kernels—such as to dot-product, polynomial, or other translation-invariant kernels—would make the findings more comprehensive and impactful.

**Questions:**

1. The feature mapping of an RKHS is not always invertible. When the feature mapping of kernel k-means is non-invertible, how does the analysis based on Bijective Centers and Order-Preserving Points still apply?
2. For kernel functions without explicit feature mappings other than the Gaussian kernel, how can their Bijective Centers be identified?
3. Line 160: What are the specific expressions for cost(T, X) and minT(T, X)?
4. The experiments are conducted only on low-dimensional and small-scale datasets. Can DT2 be applied to high-dimensional features and large-scale datasets?

---

> ### Author Response · Authors · 2025-11-19
> **Rebuttal**
>
> Thanks for your review and suggestions.
>
> # Weaknesses
>
> Our aim is not to propose a new algorithm with better interpretability, but rather to explain the price of explainability for the kernel $k$-means of the Gaussian kernel and Laplacian kernel, which explains the experimental findings in the ILCR2025 paper. Our purpose in proposing a new algorithm is also to verify this theorem.
>
> # Questions
> A1. Indeed, many kernel functions are irreversible. For these kernel functions, we can only look for order-preserving points, not Bijective Centers. This is because order-preserving points can be found using kernel tricks without requiring explicit kernel mappings.
>
> A2. For these kernel functions, we do not need to find the feature map in practice. We can use the kernel trick to find the points of the Bijective Centers if the Bijective Centers exist. Otherwise, we can only use the kernel trick to find the Order-preserving points.
>
> A3. Because various types of trees can be constructed, and the price of explainability of $X$ is $\varrho(X)$ is the cost of the tree with the minimum cost, I apologize, the correct expression should be $\varrho(\mathcal{X})=\min_\mathcal{T}\varrho(\mathcal{T}, \mathcal{X})$.
>
> A4. We used the dataset from ICLR2025. For larger and higher-dimensional datasets, since kernel k-means has a complexity of $O(n^2d)$, it is inherently difficult to scale to very large datasets. Therefore, our method is also unable to interpret large datasets that are difficult for kernel $k$-means to handle.

---

> > ### Comment · Reviewer_iUgn · 2025-11-27
> >
> > Thank you for your response. I appreciate the clarifications provided, though some of my concerns about the paper’s contribution still remain. The mention of the **“ILCR2025 paper”** also suggests there may have been an unintended reference, which could be clarified further.
> >
> > Because this work is situated within the interpretability and explainable AI track, a more thorough discussion of explanation itself—ideally starting from the preliminaries—would greatly strengthen the paper and better align it with the track’s goal of deepening our understanding of AI decision-making. I look forward to seeing future revisions that expand on these important aspects, and for now, I will keep my score unchanged.

---

### Official Review · Reviewer_sfxZ · 2025-11-06

**Soundness:** 3
**Presentation:** 3
**Contribution:** 3
**Rating:** 4
**Confidence:** 1

**Summary:**

The explainability of the machine learning model has received increasing attention recently for security and model reliability reasons. This paper studies explainable kernel clustering and compares the explainable performance of kernel k-means algorithms based on different kernels. In particular, this paper shows that kernel k-means clustering with the Laplacian kernel has a lower price of explainability than that with the Gaussian kernel. In addition, this paper proposes a new kernel k-means interpretability algorithm that directly constructs a dual-threshold tree in the original space to achieve interpretable kernel k-means, and experimentally shows that it outperforms KIMM, which constructs the threshold tree in the kernel space.

**Strengths:**

1. The structure of the paper is relatively clear, though the writing quality is slightly lacking.
2. The theoretical derivation and proof in the paper are quite thorough.
3. The interpretability of KKM represents a promising research direction with significant academic value.

**Weaknesses:**

1. There are punctuation errors and typos in the formula expressions.
2. The experimental section is excessively simplistic, and the datasets used are almost toy datasets with a tiny data scale.
3. Compared with KIMM, the performance or price improvement of the proposed method is very minimal and marginal.
4. I believed that the paper fails to adequately elaborate on how the interpretability of KKM should be explained and how it can guide kernel clustering algorithms. In terms of the paper’s contributions, the first one is not sufficient to be listed independently; the second merely involves calculations and derivations; the third is a simple algorithmic illustration. Overall, this paper significantly lacks innovation and is not qualified for publication in ICLR.

**Questions:**

As above.

---

> ### Author Response · Authors · 2025-11-19
> **Rebuttal**
>
> Thanks for your review and suggestions.
>
> # Weaknesses
>
> A1. Thank you for pointing out the typos.
>
> A2. We are using the dataset from the ICLR2025 paper. This paper is the only one that studies the price of explainability for kernel $k$-means.
>
> A3. The motivation of this paper is not to propose a new algorithm, but to theoretically prove the phenomenon found in the ICLR2025 experiment. We propose a new algorithm to verify our theorem and find that the algorithm that performs better than KIMM (our proposed algorithm) also exhibits the same phenomenon.
>
> A4. Our main contribution is the theoretical proof of the experimental findings from ICLR 2025, demonstrating that the theorem holds across different data distributions. Furthermore, for the same kernel function, different data distributions result in different interpretability prices, findings valuable to the interpretability kernel $k$-means community. Regarding innovation, as this is a theory-oriented paper, our novelty lies in being the second paper to focus on kernel $k$-means interpretability to the best of our knowledge; the first was published at ICLR 2025. We are the first to theoretically prove that the Laplacian kernel has a better interpretability price than the Gaussian kernel, consistent with the experimental findings from ICLR 2025.

---

### Meta-Review · Area_Chair_t7ha · 2026-01-06

**Summary:**

This paper presents a theory for the price of explanability for kernel k-means with Laplacian kernel and Gaussian kernel, and also gives an algorithm to verify the theory.

While the problem is interesting, this paper is not the first to study the kernel-means interpretability. The clarity of the theory needs to be improved, and empirical experiments are not convincing, as pointed out by several reviewers. The AC doesn't recommend accepting this paper.

**Reviewer Concerns:**

Concerns that were addressed by the rebuttal:
- The contribution is not clear, whether to improve the explainability or performance of kernel k-means (reviewer iUgn)




Outstanding concerns:
- Experiments are limited to simplistic and small datasets. (Reviewer sfxZ, iUgn, tfcv) No experiments are added. In fact, the authors claim that this paper is focused on theory, and there is no need for such experiments.
- The proposed algorithm is not much better than KIMM. (Reviewer sfxZ) The authors have claimed the new algorithm as a contribution, but then contradictorily justified that the algorithm is only to verify the theory.
- The theory in the paper doesn't give insights into how to design a kernel k-means clustering algorithm. (Reviewer sfxZ)
- The writing of this paper needs to be improved. The paper lacks clarity. (Reviewer iUgn, ThGX)

**Reviewer Scores:**

Reviewer sfxZ is likely to maintain or even decrease their rating. (initial score is 4)

Reviewer iUgn has indicated keeping the original score of 2.

Reviewer ThGX has indicated keeping score of 2.

Reviewer tfcv is likely to keep the original score of 6 or lower the score, considering other reviews on the theory.

---

### Decision · Program_Chairs · 2026-01-26

Reject